# GaussED: A Python Package for Sequential Experimental Design

## Abstract

Sequential algorithms are popular for experimental design, enabling emulation, optimisation and inference to be efficiently performed. For most of these applications bespoke software has been developed, but the approach is general and many of the actual computations performed in such software are identical. Motivated by the diverse problems that can in principle be solved with common code, this paper presents `GaussED`, a high-level syntax coupled to a powerful experimental design engine in `Python`, which together automate sequential experimental design for approximating a (possibly nonlinear) quantity of interest in Gaussian processes models. Using a handful of commands, `GaussED` can be used to: solve linear partial differential equations, perform tomographic reconstruction from integral data, implement Bayesian optimisation with gradient data, and emulate a complex computer model.

## 1 Introduction

This paper presents methodology and an associated `Python` package for sequential experimental design (SED). SED is often associated with computational workflows that are complicated and cumbersome, as one is required to iterate between designing an experiment (to augment a dataset with a new datum) and performing inference for a specified quantity of interest (based on the augmented dataset). Thus SED is well-placed to benefit from the development of a high-level programming syntax, implemented in a powerful and widely-used language such as `Python` (Rainforth et al., 2024). The research challenge here is to identify a class of statistical models that are sufficiently general to include important applications of SED, while being sufficiently narrow to permit both inference and SED to be efficiently and automatically performed. This paper aims to address two important open problems in the implementation of SED:

P1 automate SED for Gaussian process (GP) models with general nonlinear quantities of interest, in the setting of linear functional data (e.g. function values, gradients, integrals);

P2 circumvent the requirement for the user to specify an acquisition function for SED, enabling application to new tasks for which suitable acquisition functions are yet to be developed, in the spirit of *AutoML* (Hutter et al., 2014).

In limiting attention to the relatively narrow class of GP models in P1, we aim to develop more powerful algorithms than would have been possible in a more general-purpose framework. The setting of P1 includes SED for the important tasks of *emulating* computer models (Kennedy & O'Hagan, 2001), performing *Bayesian optimisation* (Shahriari et al., 2015), and running *probabilistic numerical methods* (Hennig et al., 2015). Bespoke packages have been developed for these individual tasks, but many of the actual computations performed in such packages are identical. Indeed, in Section 3 we demonstrate how our proposed solution can solve diverse tasks, including: solving partial differential equations using a probabilistic numerical method, performing tomographic reconstruction from integral data, implementing Bayesian optimisation with gradient data, and emulating a complex computer model. Such a solution enables advances in computational methodology to be immediately brought to bear on diverse application areas where SED is performed.

Existing packages for SED require the user to specify an *acquisition function*, which is used to select the next experiment and serves to control the exploration-exploitation trade-off (Rainforth et al., 2024). Unfortunately, the process of determining an effective acquisition function requires domain expertise and, while several choices have been documented in the literature (see e.g. Wilson et al., 2018, for acquisition functions in Bayesian optimisation), many problems that fall into the setting of P1 have not received such detailed treatment. In removing the technical burden of prescribing the acquisition function in P2, we may sacrifice a degree of performance relative to dedicated software for tasks such as Bayesian optimisation, for which bespoke acquisition functions have been developed. However, empirical results in this paper suggest that the loss of performance may be modest, and in turn we are able to considerably expand the applicability of the methodology and associated `Python` package.

## 1.1 Our Contribution

In this paper we present `GaussED`, a high-level syntax coupled to a powerful *experimental design engine* for performing SED in the nonparametric GP context. `GaussED` achieves the aims P1 and P2, just outlined. To achieve P1, and to ensure that `GaussED` can handle data arising from continuous linear functionals, we present a rigorous probabilistic treatment of conditioning for GPs. This enables us to, for example, prevent attempts to condition on a derivative that does not exist under the GP model. To achieve P2 and circumvent the user-specification of an acquisition function, we adopt a classical but surprisingly overlooked decision-theoretic approach to SED, which requires only the quantity of interest and a loss function to be specified. The loss function quantifies the loss incurred when the true quantity of interest is approximated, a notion that is meaningful in the applied context and comparatively straightforward to elicit. The computational backend for `GaussED` comprises a spectral GP, a reparametrisation trick, and stochastic optimisation over the experimental design set.

## 1.2 Related Work

Several general-purpose packages and probabilistic programming languages (PPLs) have been developed for Bayesian parameter inference in parametric models (e.g. Wood et al., 2014; Carpenter et al., 2017; Bingham et al., 2019), often based on Markov chain Monte Carlo or variational approximations in the backend. Specialised PPLs have been developed for inferring parameters that minimise a predictive loss (e.g. using neural networks; Paszke et al., 2019), often based on automatic differentiation and stochastic gradient descent. For inference in nonparametric models, specialised PPLs have been developed for GP models (e.g. Rasmussen & Nickisch, 2010; Matthews et al., 2017), including for numerical applications (ProbNum, 2021).

The combination of PPL and SED for general parametric models has received attention in Rainforth (2017, Chapter 11) and Ouyang et al. (2016); Kandasamy et al. (2018), who provided a high-level syntax for Bayesian SED. Several application-specific PPL have been also been developed for SED in parametric models (e.g. Liepe et al., 2013). The focus of much of the research involving parametric models centres around the computational challenge of conditioning random variables on observed data, a problem that is often difficult (Olmedo et al., 2018).

SED for nonparametric models has received considerable attention in the context of Bayesian optimisation; see the review of Shahriari et al. (2015). However, existing solutions are specialised to this single task. More closely related to the present paper, Paleyes et al. (2019) developed a package called `Emukit`, in which computer model emulation, Bayesian optimisation, and a number of probabilistic numerical methods are automated. However, `Emukit` focuses on function-value data as opposed to general continuous linear functionals (c.f. P1) and requires the user to specify a suitable acquisition function (c.f. P2).

Some interesting related research directions will also briefly be mentioned:

- In many scenarios there is a cost associated with each experiment, and these costs may not be equal; and important research direction is to take cost into account in SED (Zheng et al., 2020).

- There is a close relationship between *Bayesian active learning* and SED, as explored in (e.g. Gal et al., 2017; Kirsch et al., 2019).

- The vanilla approach to SED can be viewed as *myopic*, looking only one step ahead in determining the utility of an experiment. Alternative, non-myopic approaches have been developed in the context of Bayesian optimisation and could be extended to general SED (González et al., 2016; Jiang et al., 2020).

- For applications where the likelihood function is not explicit, likelihood-free algorithms for SED have been developed (e.g. Hainy et al., 2016).

- Recent researchers have aimed to cast SED as a problem in reinforcement learning (Foster et al., 2021; Blau et al., 2022; Shen & Huan, 2023); this is an interesting insight but is not an approach pursued in the current work.

For further discussion of the state-of-the-art in Bayesian experimental design we recommend the recent review of Rainforth et al. (2024).

**Outline:** The of the paper is structured as follows: Section 2 presents a detailed technical description of `GaussED`. Section 3 described the syntax of `GaussED` and presents diverse applications of SED, for which bespoke code had previously been developed but whose automation is essentially trivial using `GaussED`. The potential and limitations of `GaussED` are summarised in Section 4.

## 2 Methodology

This section presents the statistical and computational methodology used in `GaussED`. First, in Section 2.1, the notation and mathematical set-up are introduced. The elements of SED are outlined in Section 2.2 and a classical, but surprisingly overlooked, approach to SED is presented in Section 2.3. This decision-theoretic approach circumvents the requirement to specify an acquisition function and, moreover, enables state-of-the-art stochastic optimisation to be employed in SED, as explained in Sections 2.4 and 2.5. The hyperparameters of the GP model are estimated online during SED, as explained in Section 2.6.

### 2.1 Notation and Set-Up

Let $\mathcal{F}$ be a normed vector space of real-valued functions on some domain $\mathcal{X} \subseteq \mathbb{R}^d$. The problems that we consider involve a latent function $\mathsf{f} \in \mathcal{F}$, associated with a high computational cost, and the task is to approximate a (possibly nonlinear) quantity of interest $q(\mathsf{f})$ using SED. The experiments are represented[1] as continuous linear functionals $\delta : \mathcal{F} \to \mathbb{R}$ and may, for example, include pointwise evaluation $\delta(\mathsf{f}) = \mathsf{f}(x)$ of the latent function $\mathsf{f}$ at a specified location $x \in \mathcal{X}$, pointwise evaluation of a gradient, or evaluation of an integral, such as a Fourier transform. A limited computational budget motivates the careful selection of informative experiments $\delta_1, \ldots, \delta_n$. SED is often preferred[2] over *a priori* experimental design, since it allows data $\delta_1(\mathsf{f}), \ldots, \delta_{n-1}(\mathsf{f})$, which have already been observed, to inform the design of the next functional $\delta_n$.

Bayesian statistics provides a general framework for SED. Let $(\Omega, \mathcal{S}, \mathbb{P})$ be a probability space and consider a random variable $f : \Omega \to \mathcal{F}$. This serves as a statistical model for the latent $\mathsf{f}$, and encodes *a priori* knowledge, such as the smoothness of $\mathsf{f}$. To notate the distribution of $f$, we first define the *pre-image* of a set $B \subseteq \mathcal{F}$ as $f^{-1}(B) := \{\omega \in \Omega : f(\omega) \in B\}$ and we let $f_\# \mathbb{P}$ denote the *pushforward* of $\mathbb{P}$ through $f$; i.e. the probability distribution on $\mathcal{F}$ that assigns, to each Borel set $B \subseteq \mathcal{F}$, the mass $f_\# \mathbb{P}(B) := \mathbb{P}(f^{-1}(B))$. The distribution of $f$ will be denoted $\mathbb{P}_f := f_\# \mathbb{P}$ in the sequel. Our presentation allows for general priors for $f$ until Section 2.5, at which point we will assume $f$ is a GP. Throughout we adopt the convention that $\mathsf{f}$ refers to the latent function of interest, $f$ is a random variable model for $\mathsf{f}$, and $f$ is a generic element of the set $\mathcal{F}$.

---

[1] The focus of this paper is on data that are *exactly* observed. Gaussian errors can be handled in `GaussED` by building measurement error into the GP covariance model.

[2] Sequential design is known to be near-optimal under *adaptive submodularity* (Golovin & Krause, 2011).

## 2.2 Sequential Experimental Design

SED iterates between designing an experiment $\delta_n$, to augment a dataset with a new datum $\delta_n(\mathsf{f})$, and performing inference for a specified quantity of interest, based on the augmented dataset $\boldsymbol{\delta}_n(\mathsf{f}) := (\delta_1(\mathsf{f}), \ldots, \delta_n(\mathsf{f}))^\top$. Let $\mathcal{D}$ indicate the *design set*, whose elements are continuous linear functionals on $\mathcal{F}$. The design set $\mathcal{D}$ will depend on the problem at hand, and contains only the experiments that can actually be performed. At iteration $n$, SED selects an experiment $\delta_n$ from the design set in order that an *acquisition function* is maximised[3]:

$$\delta_n \in \arg\max_{\delta \in \mathcal{D}} A(\delta; \mathbb{P}_f, \boldsymbol{\delta}_{n-1}(\mathsf{f})) \tag{1}$$

The role of the acquisition function $A$ is to control the exploration-exploitation trade-off, but the computational convenience of computing (1) is also important. Much research has been dedicated to exploring choices for $A$, and the statistical and computational properties of the associated sequence $(\delta_n)_{n=1}^\infty$. Specific applications, where interest is not necessarily in $\mathsf{f}$ but rather a derived quantity of interest $q(\mathsf{f})$, have developed bespoke acquisition functions that balance computational cost with accurate approximation of the quantity of interest, in particular in Bayesian optimisation (see Table 1 in Wilson et al., 2018). This presents a major problem (P2) for the development of a *general purpose* solution to SED, since in general we cannot expect a user to specify a suitable acquisition function for the problem at hand.

As a first step toward solving P2, we consider a Bayesian approach to the design of an acquistion function. To this end, let $\mathbb{P}_f(\cdot|\boldsymbol{\delta}_n(\mathsf{f}))$ denote the conditional distribution (or *posterior*) of $f$ obtained by setting the values $\boldsymbol{\delta}_n(f)$ equal to the observed data $\boldsymbol{\delta}_n(\mathsf{f})$. From a mathematical perspective, the proper construction of a conditional distribution for an infinite-dimensional random variable $f$ is non-trivial; we suppress further discussion in the main text but refer the reader to Appendix A for full mathematical detail. A Bayesian approach to the design of an acquisition function is then to let $U : \mathbb{R}^{n-1} \times \mathbb{R} \to \mathbb{R}$ be a *utility* function, to be specified, and to seek an experiment for which the current expected utility

$$A(\delta; \mathbb{P}_f, \boldsymbol{\delta}_{n-1}(\mathsf{f})) = \int U(\boldsymbol{\delta}_{n-1}(\mathsf{f}), \delta(\mathsf{f})) \, \mathrm{d}\mathbb{P}_f(\mathsf{f}|\boldsymbol{\delta}_{n-1}(\mathsf{f})) \tag{2}$$

is maximised. The utility $U(\boldsymbol{\delta}_{n-1}(\mathsf{f}), \delta(\mathsf{f}))$ represents the value to the user of observing the datum $\delta(\mathsf{f})$. Thus the design of an acquisition function can be reduced to the design of a utility function. A popular default choice for $U$ is the *information gain* (Lindley, 1956)

$$\mathrm{KL}(\, \mathbb{P}_f(\cdot|\boldsymbol{\delta}_{n-1}(\mathsf{f}), \delta(\mathsf{f})) \parallel \mathbb{P}_f(\cdot|\boldsymbol{\delta}_{n-1}(\mathsf{f})) \,), \tag{3}$$

which quantifies the extent to which observation of the datum $\delta(\mathsf{f})$ changes *a posteriori* belief; here KL denotes the Kullback–Leibler divergence. For related approaches and discussion see the recent survey in Kleinegesse & Gutmann (2021). However, in the setting where data are exactly observed, the two distributions in (3) will be mutually singular and the Kullback–Leibler divergence will not exist. This renders information-based acquisition functions such as (3) unsuitable in our context. Instead, we revisit a classical but often overlooked idea from experimental design, next.

## 2.3 A Decision-Theoretic Approach

A general approach to construction of a utility $U$ is provided by Bayesian decision theory in the *parameter inference* context[4]. Let $L : \mathcal{F} \times \mathcal{F} \to \mathbb{R}$ denote the *loss* $L(\mathsf{f}, \mathsf{g})$ when estimating the function (or *parameter*) $\mathsf{f}$ by $\mathsf{g}$. Then we can take $U$ to be the negative Bayes' risk

$$-\min_{\mathsf{g} \in \mathcal{F}} \int L(\mathsf{g}, \mathsf{g}') \, \mathrm{d}\mathbb{P}_f(\mathsf{g}'|\boldsymbol{\delta}_{n-1}(\mathsf{f}), \delta(\mathsf{f})), \tag{4}$$

which corresponds to the negative expected loss when the Bayes act $\mathsf{g}$ is used. Compared to an acquisition function or a utility function, it can be more straightforward for a domain expert to specify a suitable loss

---

[3]To avoid pathological cases, the existence of a (not necessarily unique) maximum is always assumed.

[4]The decision-theoretic approach was advocated by Berger (1985, Section 2.5), who wrote "*better inferences can often be done with the aid of decision-theoretic machinery and inference losses*".

function $L$, since no explicit consideration of the design set, or explicit control of the exploration-exploitation trade-off, is required. Although appealing in terms of its generality, the presence of the optimisation over g has historically rendered this utility unappealing from a computational viewpoint, and motivated more convenient choices, such as (3), that have since become canonical (see the surveys in Chaloner & Verdinelli, 1995; Rainforth et al., 2024). However, we argue that the presumed intractability of loss-based utilities might need to be revisited in light of modern and powerful stochastic optimisation techniques. Indeed, for loss functions of the form $L(f, g) = \|q(f) - q(g)\|^2$, indicating that one has a quantity of interest $q(f)$ taking values in a normed space[5], under mild conditions (4) is equal to

$$-\frac{1}{2} \iint L(g, g') \, d\mathbb{P}_f(g|\boldsymbol{\delta}_{n-1}(f), \delta(f)) \, d\mathbb{P}_f(g'|\boldsymbol{\delta}_{n-1}(f), \delta(f)). \tag{5}$$

The required regularity conditions and a formal proof are contained in Appendix B. At first glance it is unclear why this observation is helpful, since we have replaced an optimisation problem with an integration problem, and integration is typically *more* difficult than optimisation. However, this formulation turns the experimental design problem to find $\delta_n$ into a double expectation and, if the design set $\mathcal{D}$ has enough structure for calculus, then gradient-based stochastic optimisation can be applied.

The restriction to squared error loss is not as limited as it may first appear, since one has the freedom to specify the quantity of interest $q(f)$ in such a way that application of squared error loss to $q(f)$ captures salient aspects of the task at hand. For example, if one is equally interested in all aspects of f then one could take $q(f) = f$, while if one is more interested in the largest values taken by f then one could take $q(f) = \exp(f)$, so that the size of $\|q(f) - q(g)\|$ is driven by the difference between the largest values taken by f and g. Concrete examples of this are provided in Section 3.2.

### 2.4 Stochastic Optimisation

Following this decision-theoretic approach, an acquisition function is obtained in expectation form by plugging (5) into (2) and applying the law of total probability, producing

$$A(\delta; \mathbb{P}_f, \boldsymbol{\delta}_{n-1}(f)) = -\frac{1}{2} \iint L(g, g') \, d\mathbb{P}_f(g'|\boldsymbol{\delta}_{n-1}(f), \delta(g)) \, d\mathbb{P}_f(g|\boldsymbol{\delta}_{n-1}(f)). \tag{6}$$

This acquisition function does not permit a closed form in general. Several numerical methods have been proposed for maximisation of acquisition functions in the literature, including Bayesian optimisation (Overstall & Woods, 2017; Kleinegesse & Gutmann, 2019), non-gradient based Monte-Carlo methods, and approximation strategies. Similar to the approach[6] of Wilson et al. (2018), here we consider the use of stochastic optimisation techniques (Robbins & Monro, 1951) for selecting an experiment $\delta$ for which (6) is approximately maximised. For an overview of stochastic optimisation, see Kushner & Yin (2003); Ruder (2016). First we perform a *reparametrisation trick* (Williams, 1992), expressing

$$g' \sim \mathbb{P}_f(\cdot|\boldsymbol{\delta}_{n-1}(f), \delta(g)) \Leftrightarrow g' = \eta(\omega; \mathbb{P}_f, \boldsymbol{\delta}_{n-1}(f), \delta(g)), \; \omega \sim \mathbb{P}, \tag{7}$$

using a deterministic transformation $\eta$ of a random variable $\omega$ that is $\delta$-independent. Section 2.5, below, details how we applied the reparametrisation trick to a GP model. Now, suppose further that the elements of the design set can be parametrised as $\mathcal{D} = \{\delta_z\}_{z \in \mathbb{R}^m} \subseteq \mathcal{F}$. Assuming sufficiently regularity for the following calculus to be well-defined, an unbiased estimator of the gradient of the acquisition function is

$$\frac{\partial}{\partial z_i} A(\delta_z; \mathbb{P}_f, \boldsymbol{\delta}_{n-1}(f)) \approx -\frac{1}{2} \frac{1}{NM} \sum_{i=1}^N \sum_{j=1}^M \frac{\partial}{\partial z_i} L(g_i, \eta(\omega_{ij}, \mathbb{P}_f, \boldsymbol{\delta}_{n-1}(f), \delta_z(g_i))),$$

---

[5] A focus on squared error loss is only a mild restriction, since we are free to re-parametrise the quantity of interest $q$ as $t \circ q$, where $t$ is an injective map (to ensure that information is not lost). Through careful selection of $t$ we may formulate the SED task in a setting where squared error loss is appropriate for the task at hand.

[6] Wilson et al. (2018) performed a reparametrisation trick by restricting attention to acquisition functions that depend on the GP only at a finite number of locations in the domain $\mathcal{X}$; in contrast, this paper exploits a spectral approximation of the GP, described in Section 2.5.

where the $g_i$ are independent random variables with distribution $\mathbb{P}_f(\cdot|\boldsymbol{\delta}_{n-1}(\mathsf{f}))$ and the $\omega_{ij}$ are independent random variables with distribution $\mathbb{P}$. This is an instance of *nested Monte Carlo*. The optimal balance between $N$ and $M$ for a fixed computational budget is discussed in Rainforth et al. (2018); for a continuously differentiable gradient, an optimal choice[7] is $N \propto M^2$. `GaussED` exploits state-of-the-art spectral GPs to perform the reparametrisation trick, as presented next.

### 2.5 Spectral Approximation of GPs

Up to this point our discussion applied to general statistical models $\mathbb{P}_f$ for the latent function $\mathsf{f}$. In the remainder GPs will be used, since they facilitate closed form conditional distributions, as appearing in (6). The purpose of this section is twofold; to briefly introduce GPs and to describe how the reparametrisation trick can be performed.

A random variable $f$ taking values in a normed vector space $\mathcal{F}$ is *Gaussian* if, for every continuous linear functional $\delta : \mathcal{F} \to \mathbb{R}$, the random variable $\delta(f)$ is a Gaussian on $\mathbb{R}$; see Definition 2.41 in Sullivan (2015). It follows that the statistical properties of a GP are characterised by its mean function $\mu(x) \coloneqq \mathbb{E}[f(x)]$, $x \in \mathcal{X}$, and covariance function $k(x, y) \coloneqq \mathbb{C}[f(x), f(y)]$, $x, y \in \mathcal{X}$, and we write $f \sim \mathcal{GP}(\mu, k)$. GPs admit conjugate inference, meaning that for a continuous linear functional $\delta \in \mathcal{D}$, the conditional distributions $\mathbb{P}_f(\cdot|\delta(\mathsf{f}))$ are also Gaussian, with mean and covariance functions that can be computed in closed form; see Appendix C.

For the reparametrisation trick, we aim to write a GP as a deterministic transformation $f = \eta(\omega)$ of a random variable $\omega$, such that the distribution of $\omega$ does not depend on $\mu$ or $k$. However, being a nonparametric statistical model, an infinite-dimensional $\omega$ will in general be required. This motivates the use of an accurate finite-dimensional approximation of a GP at the outset, i.e. for the prior $\mathbb{P}_f$. A truncated Karhunen–Loeve expansion (see e.g. Theorem 11.4 in Sullivan, 2015) in principle provides such a transformation, however this requires computation of the eigenfunctions of $k$, and linear functionals thereof, which will in general be difficult. The solution adopted in `GaussED` is to use the finite-rank approximation to isotropic GPs introduced in Solin & Särkkä (2019): $f = \eta(\omega) = \mu + \sum_{i=1}^{m} \omega_i \phi_i$, where the coefficients $\omega_i \sim \mathcal{N}(0, s(\sqrt{\lambda_i}))$ are independent, $s$ is the *spectral density* of $k$, and $(\phi_i, \lambda_i)$ are the pairs of eigenfunctions and eigenvalues of the Laplacian $\Delta$ over the domain $\mathcal{X}$; see Appendix D for detail. The approximation converges as $m \to \infty$, with small values of $m$ often practically sufficient; see Riutort-Mayol et al. (2020). `GaussED` puts the user in control of $m$, since $m$ is the principal determinant of computational complexity in the experimental design engine, aside from the computations involving the latent function $\mathsf{f}$ itself.

### 2.6 Hyperparameter Estimation

To this point we assumed that a GP model can be specified at the outset. In reality one is usually prepared only to posit a parametric class of GPs whose parameters (called *hyperparameters*) are jointly estimated. In `GaussED` the hyperparmaters of the GP are estimated at each iteration $n \geq n_0$ of SED, using the available dataset $\boldsymbol{\delta}_n(\mathsf{f})$, after an initial number $n_0 \in \mathbb{N}$ of data have been observed. Maximum likelihood estimation is employed, facilitated using automatic differentiation and Adam (Kingma & Ba, 2015). The role of $n_0$ is to guard against over-confident inferences, since maximum likelihood tends to overfit when the dataset is small; see e.g. Chapter 5 of Rasmussen & Williams (2006). In `GaussED`, the default value is taken as $n_0 = 10$.

This completes our description of `GaussED`. Our attention turns, next, to demonstrating and assessing its capabilities.

## 3 Demonstration

The aims of this section are to validate `GaussED` and to highlight the diverse and non-trivial applications that can be tackled. `GaussED` is based on `Python` and utilises the automatic differentiation capabilities of `PyTorch` (Paszke et al., 2019). Source code and documentation for `GaussED` can be downloaded from `[blinded - uploaded for review]`.

---

[7]The values $M = 9$, $N = 9^2$, were used for all experiments we report, being among the smallest values for which stochastic optimisation was routinely successful.

Full details for each of the following examples are provided in Appendix F. An investigation into the sensitivity of the computational methodology to initial conditions, the choice of stochastic optimisation method, and the number of basis functions $m$, can be found in Appendix G.

## 3.1 Probabilistic Solution of PDEs

Our first example concerns the probabilistic numerical solution of Poisson's equation with Dirichlet boundary conditions; the intention is to validate our methodology on a problem that is well-understood. SED for such problems was investigated with bespoke code in Cockayne et al. (2016). The PDE we consider is defined on $\mathcal{X} = [-1, 1]^2$ and takes the form

$$\Delta f(x) = g(x), \qquad x \in \mathcal{X},$$
$$f(x) = 0, \qquad x \in \partial\mathcal{X}.$$

Our quantity of interest is the solution f and the black-box source g is assumed to be associated with a computational

```
k = MaternKernel(3, dim=2)
qoi = SpectralGP(k)
obs = Laplacian(qoi)
loss = L2(qoi)

d = EvaluationDesign(obs, initial_design)
acq = BayesRisk(qoi, loss, d)

experiment = Experiment(obs, laplace_f, d, acq)
experiment.run(n=150)
```

Figure 1: Example syntax for `GaussED`.

cost, so that numerical uncertainty quantification is required. For this demonstration we simply took $g(x) = -320|x_1^3 \exp\{-(3.2x_1)^2 - (10x_2 - 5)^2\}|$ as a test bed. The latent f was modelled as a GP $f$ with mean zero and Matérn covariance with smoothness parameter $\nu = 3 + \frac{1}{2}$, implying that samples are contained in a normed space $\mathcal{F}$ on which the functionals $\delta(f) = \Delta f(x)$ are continuous; these functionals constitute the design set $\mathcal{D}$, parameterised by $x \in \mathcal{X}$. It is known that an optimal experimental design in this case is *space filling* (Wendland, 2004; Novak & Woźniakowski, 2010), as quantified by the *fill distance*

$$\text{FD}(\{x_i\}_{i=1}^n, \mathcal{X}) \coloneqq \sup_{x \in \mathcal{X}} \left\{ \min_{i \in \{1, \dots, n\}} \|x - x_i\| \right\},$$

and this fact will be used to validate `GaussED`. The syntax of `GaussED` is demonstrated in Figure 1, and consists of specifying a covariance function (`k`), a quantity of interest (`qoi`), an observation model (`obs`), here the Laplacian (`Laplace`), a loss function (`loss`), a design (`d`) initialised with an `initial_design`, and an acquisition function (`acq`). `BayesRisk` is the default acquisition function from (6), but `GaussED` retains the capability for alternative acquisition functions in the event that they can be user-specified. The experiment object (`experiment`) then collates these objects together to perform $n = 150$ iterations of SED, optimising hyperparameters as specified in Section 2.6.

Results are shown in Figure 2 and required only the 8 lines of code shown in Figure 1. The number of basis functions used was $m = 30^2$, we computed $n_0 = 10$ iterations of SED before beginning hyperparameter optimisation and a total of 9 CPU hours were invested to ensure that all $n = 150$ instances of stochastic optimisation converged. The convergence rate of the fill distance is lower-bounded by $\Theta(n^{-1/2})$, and Figure 2c demonstrates that this optimal rate is empirically achieved by `GaussED`. This validates our approach to SED.

## 3.2 Tomographic Reconstruction

Our next example is tomographic reconstruction from x-ray data (Mersereau & Oppenheim, 1974). The aim is to reconstruct a latent function $f : \mathcal{X} \to \mathbb{R}$, where $\mathcal{X} = [-1, 1]^d$, using line-integral data of the form

$$\delta(f) = \int_a^b f(r(t)) \, |r'(t)| \, dt,$$

where $r(t)$, $t \in [a, b]$, is a parameterisation of a line with endpoints $r(a), r(b) \in \partial\mathcal{X}$. SED for this problem was recently addressed, using bespoke code, in Burger et al. (2021) and Helin et al. (2021). Following

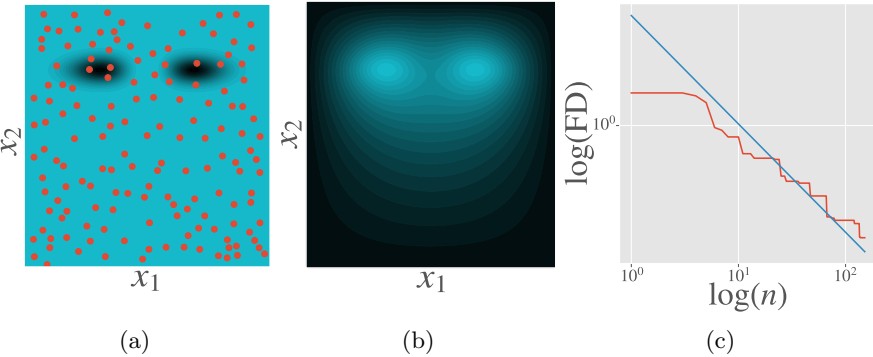

(a)            (b)            (c)

Figure 2: *Probabilistic Solution of PDEs:* (a) Source term $\mathsf{g}$ with design points (red) determined by SED overlaid. (b) Mean of $f|\boldsymbol{\delta}_n(\mathsf{f})$, the posterior obtained using SED. (c) Fill distance (FD; red) versus the number $n$ of iterations in SED, with theoretical optimal slope $-\frac{1}{2}$ (blue) displayed as a visual aid.

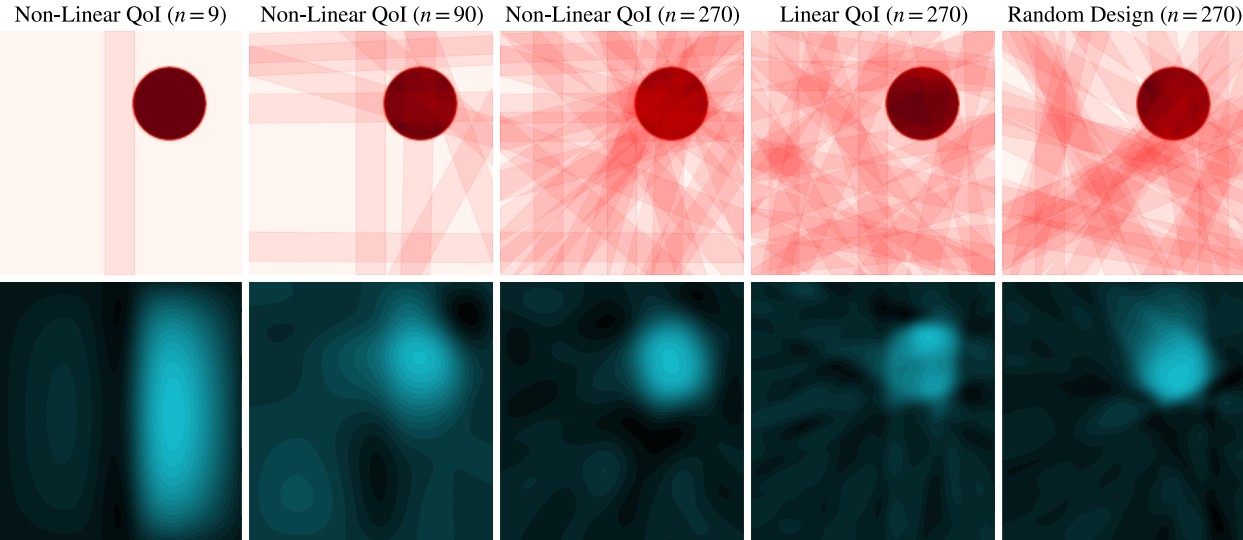

Figure 3: *Tomographic Reconstruction:* The top row displays experimental designs, overlaying the latent $\mathsf{f}$. Each red bar indicates the region over which 9 equally-spaced line integrals were computed. The bottom row displays the corresponding mean of $f|\boldsymbol{\delta}_n(\mathsf{f})$, the posterior obtained using (from left to right): SED with non-linear quantity of interest ($n = 9, 90, 270$), SED with linear quantity of interest ($n = 270$), and a random design ($n = 270$).

Burger et al. (2021), an experiment consists of a set of 9 parallel line integrals across $\mathcal{X}$, with lines a perpendicular distance of 0.03 apart. As a toy example, we consider tomographic reconstruction of an indicator function $\mathsf{f}(x) = \mathbb{1}_B(x)$ where $B$ is the ball of radius 0.3 centred on $(0.4, 0.4)$. For our statistical model $f$ we used a stationary GP with Matérn covariance and smoothness parameter $\nu = 2 + \frac{1}{2}$, and the non-linear quantity of interest was $q(\mathsf{f}) = \exp(3\mathsf{f})$ which, when combined with squared error loss, serves to prioritise the reconstruction of the ball in SED. See Appendix F.2 for full detail.

Results are shown in Figure 3 and only 32 lines of code were required. In this experiment, we used $m = 28^2$ basis functions and began optimising hyperparameters at SED iteration $n = 1$. In total, 2.5 CPU hours were required. SED using `GaussED` provides improved reconstruction compared to a random design (right panel). As an additional comparison, we also performed SED with the linear quantity of interest $q(\mathsf{f}) = \mathsf{f}$ and a space-filling design was obtained. Exploratory investigation of this kind is straight-forward in `GaussED`.

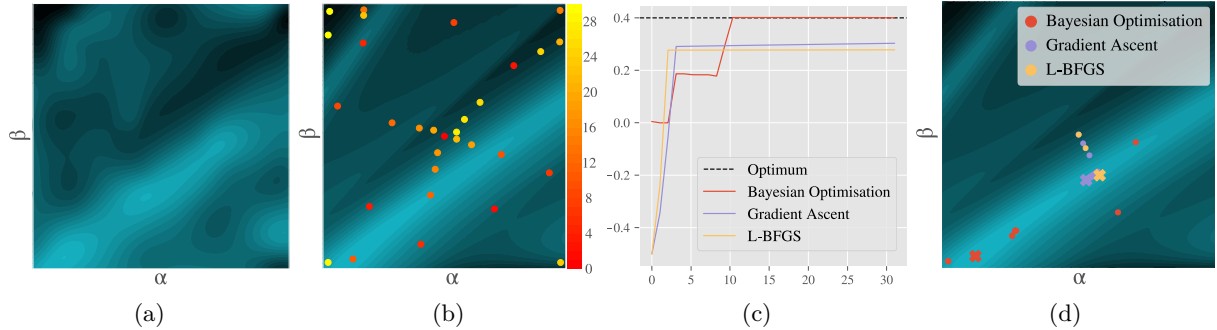

Figure 4: *Gradient-Based Bayesian Optimisation:* (a) Mean of $f|\boldsymbol{\delta}_n(\mathsf{f})$, the posterior after $n = 90$ total evaluations. (b) Log-likelihood $\mathsf{f}$, with design points overlaid. Colour indicates the order in which points were selected in SED. (c) Maximum value of the likelihood obtained during the first $m$ iterations of each optimisation method. (d) Location of the maximum value along the optimisation path, where the colored ✖ symbols indicate the maximum value obtained (for Bayesian optimisation, the maximum of the posterior mean is reported).

### 3.3   Gradient-Based Bayesian Optimisation

Our next example uses Bayesian optimisation to perform parameter inference via maximum likelihood, and for this we consider the Lotka–Volterra model

$$\begin{aligned}
\frac{\mathrm{d}p}{\mathrm{d}t} &= \alpha p - \beta pq, \\
\frac{\mathrm{d}q}{\mathrm{d}t} &= -\gamma q + \delta pq,
\end{aligned} \tag{8}$$

where $p(t), q(t) > 0$ are the predator and prey populations, respectively, at time $t$ and $\alpha, \beta, \gamma$ and $\delta$ are free parameters to be inferred. To facilitate visualisation of experimental designs we consider inferring only $\alpha$ and $\beta$, which we collect in a single parameter vector $x = (\alpha, \beta)$. For this demonstration we restrict attention to $\mathcal{X} = [0.45, 0.9] \times [0.09, 0.5]$, to avoid failure of the numerical integrator applied to (8). The remaining parameters, $\gamma$ and $\delta$, are then taken as fixed. Our latent function $\mathsf{f}$ is the log-likelihood, denoted $\mathsf{f} = \log \mathcal{L}$, arising from a particular dataset of noise-corrupted observations described in Appendix F.3. Our quantity of interest is the maximum likelihood estimator $q(\mathsf{f}) = \max_{x \in \mathcal{X}} \mathsf{f}(x)$. The design set $\mathcal{D}$ contains pointwise evaluation functionals $\delta_x^1(\mathsf{f}) = \log \mathcal{L}(x)$ and gradient evaluation functionals $\delta_x^{2,i}(\mathsf{f}) = \nabla_{x_i} \log \mathcal{L}(x)$, and at each iteration of SED we evaluate $(\delta_x^1(\mathsf{f}), \delta_x^{2,1}(\mathsf{f}), \delta_x^{2,2}(\mathsf{f}))$ for some $x \in \mathcal{X}$, mimicking the information provided when (8) is solved using an adjoint method. Through a suitable sequence of evaluation functionals, SED aims to approximate the maximum likelihood estimator.

Results are shown in Figure 4 and only 17 lines of code were required. In this experiment we used $m = 35^2$ basis functions, we computed $n_0 = 10$ iterations of SED before beginning hyperparameter optimisation and 1.5 CPU hours were required. For reference, results based on gradient ascent and L-BFGS (Nocedal, 1980) are also displayed. All algorithms were initialised at the midpoint of the domain $\mathcal{X}$ and run for $n = 30$ iterations. Bayesian optimisation with gradient data outperformed the first order optimisation methods in this example, where attention is focused on performance after a small number of likelihood evaluations, to mimic more challenging applications in which the likelihood is associated with a more substantial computational cost. In Appendix G.3, we compare our general form of an acquisition function against a bespoke acquisition function used in Bayesian optimisation; *expected improvement* (Jones et al., 1998). These additional results suggest that the performance gap between the default acquisition function used in `GaussED` and bespoke acquisition functions, tailored for specific tasks, may not be substantial.

### 3.4 Emulation of a Cardiac Model

Our final example is a realistic use-case of `GaussED`; emulating a complex biomechanical cardiac model. The model $f : \mathcal{X} \to \mathbb{R}^{25}$, due to Davies et al. (2019), comprises a system of partial differential equations (PDEs), depending on a 4-dimensional input parameter $x$ taking values in $\mathcal{X} = [0.1, 5]^4$. The model outputs quantities describing the heartbeat, and there is clinical interest in inferring input parameters $x$ for which $f(x)$ is consistent with the corresponding physical quantities measured from an MRI scan of a patient.

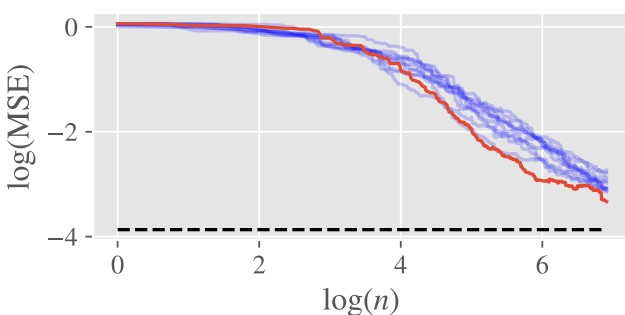

Figure 5: *Emulation of a Cardiac Model:* Plot of MSE after $n$ iterations of SED (red) and MSE obtained from conditioning on all $10^4$ datapoints (dashed). Blue lines are MSE using random evaluation points.

The complexity of the finite-element methods used to discretise the PDEs, combined with the dimension of the parameter space, mean that weeks of computation can be required to produce clinically relevant output. GP emulation seeks to reduce this computational cost, by replacing the simulator with a surrogate model trained on a small number of simulations from the original model (Kennedy & O'Hagan, 2001). Noè et al. (2019) constructed a GP emulator of the cardiac model $f$ using simulations $\delta_i(f) = f(x_i)$ at an *a priori* experimental design based on Sobol points $x_1, \ldots, x_N$, $N = 10^4$, scaled and shifted to $\mathcal{X}$. Our aim is to investigate whether computational resources can be saved using SED.

Results are shown in Figure 5 and required only the 29 lines of code contained in Appendix F.4. The number of basis functions used was $m = 442$, we first performed $n_0 = 100$ iterations of SED before beginning hyperparameter optimisation and a total of 24 CPU hours were required. To avoid re-running the cardiac model, we restricted SED to the discrete set $\{x_1, \ldots, x_N\}$ of values reported in Noè et al. (2019), but in reality the resources required to evaluate $f$ dwarf those required for SED. Emulator performance was quantified using held-out mean square error (MSE) computed on a test set of size $10^2$. For demonstration purposes, we considered just one output (the lower ventricle chamber volume) from the cardiac model. SED (red) is seen to out-perform random designs (blue) in Figure 5. Moreover, the MSE after $10^3$ iterations of SED was 0.0366, which reduces only to 0.0209 using all $10^4$ simulations from the model. Thus MSE within a factor of 2 compared to that of Noè et al. (2019) was obtained using SED based on only $10^3 \ll 10^4$ simulations from the cardiac model, highlighting both the utility of SED and the convenience of a PPL such as `GaussED` in this context.

## 4 Discussion

This paper introduced `GaussED`, a high-level syntax coupled to a powerful engine for SED. Through four experiments we illustrated the diverse applications that can be automatically solved using `GaussED`. However, automation of SED comes at a cost: Firstly, `GaussED` is restricted both to continuous linear functional data and to GPs, limiting the potential for more flexible statistical models to be employed. Alternative solutions, such as `Emukit`, offer more modelling flexibility but require acquistion functions to be manually specified. Secondly, in automating the specification of an acquisition function in `GaussED`, there may be a loss in performance terms compared to bespoke solutions for specific tasks. Our experiments involving Bayesian optimisation in Section 3.3 were encouraging, however, and suggested that such performance gaps, if they do exist, may be acceptably small. One role for `GaussED` in these settings is to provide an off-the-shelf benchmark for SED, against which more sophisticated methods can be compared.

### Acknowledgments

The authors are grateful for constructive feedback from three expert Reviewers.

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

## Appendices

Appendix A contains the mathematical preliminaries for the subsequent sections Appendix B and Appendix C. Appendix B presents the conditions for the equivalence of (4) and (5) as advertised in Section 2.3. Appendix C presents the formal background of conditioning on continuous linear data for Gaussian processes and presents properties of the Matérn covariance function. Appendix D presents a derivation of the Gaussian process model that forms the foundation of `GaussED`. Appendix E discusses computational aspects of `GaussED`. In particular, we discuss linear algebra solvers, different approaches to sampling from the posterior, and we present a complete description of how `GaussED` attempts to optimise the acquisition function in SED. Appendix F contains full details of the experiments presented in Section 3. Appendix G presents further empirical evaluation of `GaussED`.

## A    Mathematical Preliminaries

In this section we present the mathematics required to ensure that the conditioning of stochastic processes in the main text is well-defined (Appendix A.1), as well as recalling the concept of a Fréchet derivative (Appendix A.2).

### A.1    Conditioning as Disintegration

In finite dimensions, conditioning of random variables can be performed using the density formulation of Bayes' theorem. However, typical stochastic processes will be infinite-dimensional, meaning that (Lebesgue) densities do not exist in general. This necessitates a level of mathematical abstraction to ensure that conditional probabilities are well-defined. The appropriate notion, for this work, is that of *disintegration*, defined next.

Let $(\mathcal{F}, \mathcal{S}_\mathcal{F})$ and $(\mathcal{Y}, \mathcal{S}_\mathcal{Y})$ be measurable spaces and let $\delta$ be a measurable function from $\mathcal{F}$ to $\mathcal{Y}$. Recall that $\delta^{-1}(S) = \{f \in \mathcal{F} : \delta(f) \in S\}$ denotes the pre-image of $S \in \mathcal{S}_\mathcal{Y}$. Let $\mathbb{P}$ be a probability measure on $(\mathcal{F}, \mathcal{S}_\mathcal{F})$ and recall that $\delta_\#\mathbb{P}$ denotes the *pushforward* measure $(\delta_\#\mathbb{P})(S) := \mathbb{P}(\delta^{-1}(S))$ on $\mathcal{Y}$.

**Definition 1.** *The collection* $\{\mathbb{P}(\cdot|y)\}_{y\in\mathcal{Y}}$ *is called a* $\delta$-disintegration *of* $\mathbb{P}$ *if*

> *1.* $\mathbb{P}(\delta^{-1}(y)|y) = 1$ *for* $\delta_\#\mathbb{P}$ *almost all* $y \in \mathcal{Y}$

*and, for each measurable function* $g : \mathcal{F} \to [0, \infty)$*, we have*

> *2.* $y \mapsto \int g(f)d\mathbb{P}(f|y)$ *is measurable*

> *3.* $\int g(f)d\mathbb{P}(f) = \int \int g(f)d\mathbb{P}(f|y)d\delta_\#\mathbb{P}(y)$

A disintegration is a particular instance of a *regular conditional distribution* which also satisfies property (1) in Definition 1; see Chang & Pollard (1997). A basic theorem on the existence and $\delta_\#\mathbb{P}$ almost everywhere uniqueness of disintegrations is given in Parthasarathy (2005, p147). Two disintegrations will be identified if they coincide $\delta_\#\mathbb{P}$ almost everywhere, and we will therefore refer to *the* $\delta$-disintegration of $\mathbb{P}$. The concept of disintegration makes precise what it means to "condition GPs on data", and imposes conditions on the linear functionals in the design set as explained in Appendix C.

### A.2    Fréchet Derivatives

Recall that $\mathcal{F}$ was defined as a normed vector space, meaning that the notion of a *Fréchet derivative* can be exploited. A function $q : \mathcal{F} \to \mathbb{R}^d$ is called *Fréchet differentiable* at $f \in \mathcal{F}$ if there exists a bounded linear operator $A : \mathcal{F} \to \mathbb{R}^d$ such that

$$\lim_{\|g\|\to 0} \frac{\|q(f+g) - q(f) - A(g)\|}{\|g\|} = 0.$$

If such an operator exists it can be shown to be unique, called the *Fréchet derivative* of $q$ at f, and denoted $Dq(f) = A$. To emphasise that the Fréchet derivative is an operator, we occasionally write $Dq(f)(\cdot)$ in the sequel. A Fréchet derivative $Dq(f)$ is said to have *full rank* if $Dq(f)(g) = 0$ implies $g = 0$.

The chain rule for Fréchet derivatives takes the form

$$D(b \circ a)(f)(\cdot) = (Db \circ a)(f) \circ Da(f)(\cdot).$$

As a concrete example, that we use later, consider $a(f) = q(f)$ to be the quantity of interest and $b(x) = \|x - q(g)\|^2$ for all $x \in \mathbb{R}^d$ and some fixed $g \in \mathcal{F}$. Then $Db(x)(\cdot) = 2\langle x - q(g), \cdot \rangle$ is a linear operator from $\mathbb{R}^d$ to $\mathbb{R}$ and we have

$$D(b \circ a)(f)(\cdot) = 2\langle q(f) - q(g), Dq(f)(\cdot) \rangle, \tag{9}$$

which is a linear operator from $\mathcal{F}$ to $\mathbb{R}$. Further background on Fréchet derivatives can be found in Berger (1977, Section 2.1C).

An important technical result on Fréchet derivatives, that we will use in the sequel, is when the interchange of a Fréchet derivative and an integral can be permitted:

**Proposition 1.** *Let $\mathcal{F}$ be complete (i.e. a Banach space) and $(\Omega, \mathcal{S}, \mathbb{P})$ be a probability space. Let $\ell : \mathcal{F} \times \Omega \to \mathbb{R}$ satisfy the following:*

1. *f $\mapsto \ell(f, \omega)$ is Fréchet differentiable, for each $\omega \in \Omega$*

2. *$\omega \mapsto \ell(f, \omega)$ is integrable, for each $f \in \mathcal{F}$*

3. *$\omega \mapsto D\ell(f, \omega)(g)$ is integrable, for each $f, g \in \mathcal{F}$*

4. *$\int \|D\ell(f, \omega)\| d\mathbb{P}(\omega) < \infty$*

*Then the function*

$$r(f) := \int \ell(f, \omega) d\mathbb{P}(\omega)$$

*is Fréchet differentiable, with derivative*

$$Dr(f)(\cdot) = \int D\ell(f, \omega)(\cdot) d\mathbb{P}(\omega).$$

*Proof.* A special case of Kammar (2016). □

## B  Regularity Conditions for the Decision Theoretic Formulation

The aim in this section is to establish sufficient conditions for the equivalence of (4) and (5) as advertised in Section 2.3. To achieve this, we will use the notion of a Fréchet derivative from Appendix A.2. Our sufficient conditions are presented in Appendix B.1. A short discussion of the strength of these conditions is contained in Appendix B.2.

### B.1  From Optimisation to Expectation

Firstly, we rigorously establish an infinite-dimensional analogue of the classical result that the posterior mean is a Bayes act for squared error loss:

**Proposition 2.** *Let $L(f, g) = \|q(f) - q(g)\|^2$. Assume that $\mathcal{F}$ is complete (i.e. a Banach space) and that:*

(A1)  *$q : \mathcal{F} \to \mathbb{R}^d$ is Fréchet differentiable;*

(A2)  *the Fréchet derivative $Dq(f)$ has full rank at all $f \in \mathcal{F}$;*

(A3) $\int \|q(\mathrm{g})\|^2 \mathrm{d}\mathbb{P}_f(\mathrm{g}|\boldsymbol{\delta}_n(\mathrm{f})) < \infty$.

*Then any solution to*

$$\underset{\mathrm{f}\in\mathcal{F}}{\arg\min}\, r(\mathrm{f}), \qquad r(\mathrm{f}) := \int L(\mathrm{f},\mathrm{g})\ \mathrm{d}\mathbb{P}_f(\mathrm{g}|\boldsymbol{\delta}_n(\mathrm{f})) \tag{10}$$

*satisfies*

$$q(\mathrm{f}) = \int q(\mathrm{g})\ \mathrm{d}\mathbb{P}_f(\mathrm{g}|\boldsymbol{\delta}_n(\mathrm{f})).$$

*Proof.* From an application of Proposition 1 with $\ell(\mathrm{f},\omega) = L(\mathrm{f}, g(\omega))$, where $g : \Omega \to \mathcal{F}$ is a random variable with distribution $\mathbb{P}_f(\cdot|\boldsymbol{\delta}_n(\mathrm{f}))$, we deduce that our assumptions on $L$ and $q$ (A1) are sufficient for the Fréchet derivative of $r$ to exist. Thus, a minimiser f of (10) satisfies $\mathrm{D}r(\mathrm{f}) = 0$. To evaluate $\mathrm{D}r$ we exploit the integrability assumption (A3) on $q$ to differentiate under the integral, which is also justified from Proposition 1:

$$\mathrm{D}r(\mathrm{f})(\cdot) = \int \mathrm{D}L(\mathrm{f},\mathrm{g})\ \mathbb{P}_f(\mathrm{g}|\boldsymbol{\delta}_n(\mathrm{f})).$$

Next we apply the chain rule for Fréchet derivatives in the form of (9), yielding

$$\mathrm{D}r(\mathrm{f})(\cdot) = \int 2\langle q(\mathrm{f}) - q(\mathrm{g}), \mathrm{D}q(\mathrm{f})(\cdot)\rangle\ \mathrm{d}\mathbb{P}_f(\mathrm{g}|\boldsymbol{\delta}_n(\mathrm{f}))$$

$$= 2\left\langle \underbrace{q(\mathrm{f}) - \int q(\mathrm{g})\mathrm{d}\mathbb{P}_f(\mathrm{g}|\boldsymbol{\delta}_n(\mathrm{f}))}_{(*)}, \mathrm{D}q(\mathrm{f})(\cdot)\right\rangle.$$

Since $\mathrm{D}q(\mathrm{f})$ was assumed to have full rank (A2), if $\mathrm{D}r(\mathrm{f}) = 0$ then $(*) = 0$, whence the claimed result. $\square$

Now we are able to prove the advertised result:

**Proposition 3.** *In the setting of Proposition 2, and under assumptions* (A1-3), *we have*

$$\underset{\mathrm{f}\in\mathcal{F}}{\min}\, r(\mathrm{f}) = \frac{1}{2}\iint L(\mathrm{g},\mathrm{g}')\ \mathrm{d}\mathbb{P}_f(\mathrm{g}|\boldsymbol{\delta}_n(\mathrm{f}))\ \mathrm{d}\mathbb{P}_f(\mathrm{g}'|\boldsymbol{\delta}_n(\mathrm{f})).$$

*Proof.* Let $\mathrm{f} \in \mathcal{F}$ solve (10). Then consider the algebraic identity

$$q(\mathrm{g}) - q(\mathrm{g}') = \{q(\mathrm{g}) - q(\mathrm{f})\} - \{q(\mathrm{g}') - q(\mathrm{f})\}.$$

Using this identity, the loss function can be expressed as

$$L(\mathrm{g},\mathrm{g}') = \|q(\mathrm{g}) - q(\mathrm{g}')\|^2$$
$$= \|q(\mathrm{g}) - q(\mathrm{f})\|^2 - 2\langle q(\mathrm{g}) - q(\mathrm{f}), q(\mathrm{g}') - q(\mathrm{f})\rangle + \|q(\mathrm{g}') - q(\mathrm{f})\|^2.$$

From linearity of the inner product we have that

$$\iint \langle q(\mathrm{g}) - q(\mathrm{f}), q(\mathrm{g}') - q(\mathrm{f})\rangle\ \mathrm{d}\mathbb{P}_f(\mathrm{g}|\boldsymbol{\delta}_n(\mathrm{f}))\ \mathrm{d}\mathbb{P}_f(\mathrm{g}'|\boldsymbol{\delta}_n(\mathrm{f}))$$

$$= \left\langle \underbrace{\int q(\mathrm{g}) - q(\mathrm{f})\ \mathrm{d}\mathbb{P}_f(\mathrm{g}|\boldsymbol{\delta}_n(\mathrm{f}))}_{=0}, \underbrace{\int q(\mathrm{g}') - q(\mathrm{f})\ \mathrm{d}\mathbb{P}_f(\mathrm{g}'|\boldsymbol{\delta}_n(\mathrm{f}))}_{=0} \right\rangle = 0,$$

where we have used the integrability assumption (A3) on $q$ to bring the integrals into the inner product, and we have used Proposition 2 to conclude that each argument is equal to 0. Finally, from the fact that $g$ and $g'$ are identically distributed, we have

$$
\frac{1}{2} \iint L(g, g') \, d\mathbb{P}_f(g|\boldsymbol{\delta}_n(f)) \, d\mathbb{P}_f(g'|\boldsymbol{\delta}_n(f))
$$
$$
= \frac{1}{2} \iint \|q(g) - q(f)\|^2 + \|q(g') - q(f)\|^2 \, d\mathbb{P}_f(g|\boldsymbol{\delta}_n(f)) \, d\mathbb{P}_f(g'|\boldsymbol{\delta}_n(f))
$$
$$
= \frac{1}{2} \times 2 \times \int \|q(g) - q(f)\|^2 \, d\mathbb{P}_f(g|\boldsymbol{\delta}_n(f)) = \min_{f \in \mathcal{F}} r(f),
$$

which completes the argument. $\qquad\square$

## B.2 Verifying the Assumptions

The main assumption in Proposition 2 is (A2); the requirement that $Dq(f)$ has full rank for all $f \in \mathcal{F}$. As we explain below through a worked example, (A2) is non-trivial but may often be satisfied with only minor modification to the SED task in hand.

As a worked example, suppose $\mathcal{F}$ is a Hilbert space containing smooth, real-valued functions defined on a compact set $\mathcal{X} \subset \mathbb{R}^d$. Suppose that we are interested in the quantity of interest

$$
q(f) = \int_{\mathcal{X}} f(x) dx. \tag{11}
$$

Then (A2) is *not* satisfied in general, because $q(f + g) = q(f)$ for all $g$ in the linear subspace $\mathcal{G} = \{g \in \mathcal{F} : \int_{\mathcal{X}} g(x) dx = 0\}$. It follows that $Dq(f)(g) = 0$ for all $g \in \mathcal{G}$, so that $Dq(f)$ does not have full rank whenever $\mathcal{G}$ is non-trivial. However, (A2) *is* satisfied if we restrict attention to the normed vector space $\mathcal{F}_c$ spanned by the elements of $\mathcal{F} \setminus \mathcal{G}$, since then $Dq(f)(g) = \int_{\mathcal{X}} g(x) dx$ and thus $Dq(f)(g) = 0$ with $g \in \mathcal{F}_c$ implies $g = 0$. This illustrates that, with a small amount of technical care, the assumptions of Proposition 2 can often be satisfied.

## C Properties of Gaussian Processes

In this section, we present the formal background of conditioning on continuous linear data for Gaussian processes.

Let $\mathcal{X}$ be a compact subset of $\mathbb{R}^d$ for some $d \in \mathbb{N}$ and let $C^r(\mathcal{X})$ denote the vector space of $r$-times continuously differentiable real-valued functions on $\mathcal{X}$ equipped with the norm

$$
\|f\|_{C^r(\mathcal{X})} = \max_{|\alpha| \leq r} \|f^{(\alpha)}\|_\infty,
$$

where the maximum ranges over multi-indices $\alpha \in \mathbb{N}_0^d$ with $|\alpha| = \alpha_1 + \cdots + \alpha_d \leq r$ and $f^{(\alpha)}(x) := \partial_{x_1}^{\alpha_1} \ldots \partial_{x_d}^{\alpha_d} f(x)$. In what follows we consider disintegration in the case where $\mathcal{F} = C^r(\mathcal{X})$, equipped with the Borel $\sigma$-algebra, and $\mathcal{Y} = \mathbb{R}$. For an operator $\delta$ and a bivariate function $k(\cdot, \cdot)$, denote $\delta k(\cdot, \cdot)$ to be the action of $\delta$ on the first argument of $k$, and denote $\bar{\delta} k(\cdot, \cdot)$ to be the action of $\delta$ on the second argument of $k$.

**Lemma 1.** *Let $\mathbb{P}$ be a Gaussian measure on $C^r(\mathcal{X})$ with mean function $m : \mathcal{X} \to \mathbb{R}$ and covariance function $k : \mathcal{X} \times \mathcal{X} \to \mathbb{R}$. Let $\delta : C^r(\mathcal{X}) \to \mathbb{R}$ be a continuous linear functional. For each $y \in \mathbb{R}$, define $\mathbb{P}(\cdot|y)$ to be a Gaussian measure with mean and covariance function*

$$
m_y(x) = m(x) + [\bar{\delta} k(x, \cdot)][\delta \bar{\delta} k(\cdot, \cdot)]^{-1}(y - m(x))
$$
$$
k_y(x, x') = k(x, x') - [\bar{\delta} k(x, \cdot)][\delta \bar{\delta} k(\cdot, \cdot)]^{-1}[\delta k(\cdot, x')].
$$

*Then $\{\mathbb{P}(\cdot|y)\}_{y \in \mathbb{R}}$ is a $\delta$-disintegration of $\mathbb{P}$.*

*Proof.* The proof is by direct verification of properties (1-3) in Definition 1. See e.g. p.188 of Ritter (2007). □

The fact that the elements $\mathbb{P}(\cdot|y)$ of the disintegration are again Gaussian enables the repeated application of Lemma 1, for example to condition on $n \geq 1$ continuous linear functionals $\boldsymbol{\delta}_n = (\delta_1, \ldots, \delta_n)^\top$, as exploited in the main text. Constructed in this way, it can be verified that the elements $\mathbb{P}(\cdot|\boldsymbol{y}_n)$ of the resulting disintegration, with $\boldsymbol{y}_n \in \mathcal{Y}^n$, are invariant to the order in which the disintegrations are performed.

# D    Spectral Approximation

This section presents an informal derivation of the spectral GP approximation of Solin & Särkkä (2019). The following utilises properties of the Fourier transform, which are first briefly recalled.

## D.1    Properties of the Fourier Transform

In the following we use $F$ to denote the Fourier transform operator and use the notation $\hat{f} := F(f)$ to denote the Fourier transform of $f$. In the following we use the convention of using the angular frequency domain. Therefore, for square-integrable $f : \mathbb{R}^d \to \mathbb{R}$, we have

$$F(f) = \frac{1}{(2\pi)^d} \int f(x) \exp(i\langle \omega, x \rangle) \, dx.$$

Recall that, when an operator $T$ satisfies $F(Tf)(\omega) = m(\omega)\hat{f}(\omega)$, the operator $T$ is called a *multiplier operator* and the corresponding $m$ is called the *multiplier* of $T$. As a trivial example, the identity operator $Tf = f$ is a multiplication operator, with associated multiplier 1. A more elaborate example, that is used in the subsequent section, is the Laplace operator $\Delta := \frac{\partial^2}{\partial x_1^2} + \ldots + \frac{\partial^2}{\partial x_d^2}$, acting on twice differentiable functions $f : \mathbb{R}^d \to \mathbb{R}$. It can be shown that

$$F(\Delta f) = -\|\omega\|^2 F(f). \tag{12}$$

Therefore, the Laplace operator is a multiplier operator with corresponding multiplier $-\|\omega\|^2$. Similarly, compositions of Laplace operators $\Delta^n := \underbrace{\Delta \circ \ldots \circ \Delta}_{n \text{ times}}$, acting on sufficiently smooth functions $f$, is also a multiplier operator with multiplier $(-\|\omega\|^2)^n$. This can be seen by induction on the previous formula,

$$F(\Delta^n f) = -\|\omega\|^2 F(\Delta^{n-1} f) = \ldots = (-\|\omega\|^2)^n F(f).$$

By the convolution theorem, every multiplier operator $T$ with multiplier $m_T$, has an associated convolution kernel $k_T := F^{-1}(m_T)$ that satisfies the following

$$F(Tf)(\omega) = m_T(\omega)\hat{f}(\omega)$$
$$Tf = F^{-1}(m_T \hat{f}) = f \star F^{-1}(m_T) = f \star k_T,$$

where $\star$ denotes convolution. Thus a multiplier operator is, in this sense, equivalent to a convolution operation.

We now state two important results that define the intimate connection between covariance functions and the Fourier transform. The first result is known as *Bochner's theorem* (Rudin, 1990).

**Theorem 1** (Bochner's theorem). *A stationary covariance function, i.e. a covariance function of the form $k(x, y) = k(x - y)$, $k : \mathbb{R}^d \to \mathbb{R}$, can be written as the inverse Fourier transform of a finite positive measure $\mu$ such that $k(0) = \mu(\mathbb{R}^d)$. That is*

$$k(x) = \frac{1}{(2\pi)^d} \int \exp\left(i\langle \omega, x \rangle\right) \, d\mu(\omega).$$

The measure $\mu$ is called the *spectral measure* of $k$ and the density of $\mu$, if it exists, is called the *spectral density* $s(\omega)$ of $k$. In the case where the spectral density $s(\omega)$ of a stationary covariance function $k$ exists, $k$ and $s$ exist as Fourier duals. This result is known as the *Wiener–Khintchine theorem* (Khintchine, 1934).

**Theorem 2** (Wiener–Khintchine theorem). *Suppose that the spectral density $s : \mathbb{R}^d \to \mathbb{R}$ of a stationary covariance function $k : \mathbb{R}^d \to \mathbb{R}$ exists, then*

$$k(x) = \frac{1}{(2\pi)^d} \int s(\omega) \exp\left(i\langle \omega, x \rangle\right) \, \mathrm{d}\omega, \qquad s(\omega) = \int k(x) \exp\left(-i\langle \omega, x \rangle\right) \, \mathrm{d}s.$$

In the proceeding section the Wiener–Khintchine theorem and the equivalence between a multiplier operator and an associated convolution operation are both used to establish a correspondence between the covariance operator of a stationary kernel $k$ and its spectral density $s$. This is the foundation upon which the spectral Gaussian process approximation of Solin & Särkkä (2019) is established.

## D.2 Spectral Gaussian Processes

For every covariance function $k$, there exists an associated Hilbert–Schmidt integral operator, termed the *covariance operator*,

$$\mathcal{K}f = \int k(\cdot, y) f(y) \, \mathrm{d}y.$$

When $k$ is stationary, the resulting covariance operator takes the form of a convolution

$$\mathcal{K}f(x) = \int k(x - y) f(y) \, \mathrm{d}y = (f \star k)(x).$$

By the convolution theorem, we can then write the operator in the form $F(\mathcal{K}f) = \hat{k}\hat{f}$ and so $\mathcal{K}$ is a multiplier operator with multiplier $\hat{k}$. By Theorem 2, the multiplier of $\mathcal{K}$ is the spectral density $s = \hat{k}$ of $k$.

Assuming now that the covariance function is isotropic and so satisfies

$$k(x, y) = k(\|x - y\|),$$

the corresponding spectral density $s$ of $k$ can be written as a function of $\|\omega\|$ only and so $s(\omega) = S(\|\omega\|)$, for an appropriate function $S$. As a further manipulation, we can write $s$ as a function of $\|\omega\|^2$ only, $s(\omega) = \psi(\|\omega\|^2)$. Assuming that $\psi$ possesses a Taylor expansion, we can write

$$s(\omega) = \psi(\|\omega\|^2) = \sum_{i=0}^{\infty} \mu_i (\|\omega\|^2)^i,$$

with each $\mu_i \in \mathbb{R}$. Inspired by the multiplier $-\|\omega\|^2$ of the Laplacian in (12) and by utilising the above Taylor expansion, we can write the Fourier transform of the covariance operator of an isotropic kernel in the form

$$F(\mathcal{K}f)(\omega) = s(\omega)\hat{f}(\omega) = \sum_{i=0}^{\infty} \mu_i (\|\omega\|^2)^i \hat{f}(\omega) = \sum_{i=0}^{\infty} \mu_i F((-\Delta)^i f).$$

By continuity of $F$, taking the inverse Fourier transform of the above yields a polynomial expansion form of the covariance operator

$$\mathcal{K}f = \sum_{i=0}^{\infty} \mu_i (-\Delta)^i f. \tag{13}$$

The remaining step is to approximate the negative Laplacian operator. To achieve this, we write the convolution kernel $k_{-\Delta}$ of the negative Laplacian as a Mercer expansion. To this end, we consider the following eigenvalue problem of the Laplacian over a compact domain $\mathcal{X} \subseteq \mathbb{R}^d$, with boundary $\partial\mathcal{X}$, with Dirichlet boundary conditions

$$-\Delta\phi_i(x) = \lambda_i \phi_i(x), \quad x \in \mathcal{X}, \tag{14}$$

$$\phi_i(x) = 0, \qquad x \in \partial\mathcal{X}. \tag{15}$$

Over a suitable domain contained within $L^2(\mathcal{X})$, the negative Laplacian is a positive definite Hermitian operator and so we can provide a Mercer expansion of the convolution kernel $k_{-\Delta}$ of the negative Laplacian, utilising the eigenfunctions $\phi_i$. Similarly, we can provide a Mercer expansion of the convolution kernel of $(-\Delta)^n$, noting that each $\phi_i$ is again an eigenfunction, but now with corresponding eigenvalue $\lambda_i^n$. This can be seen by iteratively applying $-\Delta$ to the eigenvalue problem (14). Therefore, we have

$$(-\Delta)^n f(x) = f \star k_{(-\Delta)^n}(x) = \int k_{(-\Delta)^n}(x-y)f(y)\,\mathrm{d}y,$$

where

$$k_{(-\Delta)^n}(x-y) = \sum_{j=1}^{\infty} \lambda_j^n \phi_j(x)\phi_j(y).$$

Plugging the preceding formula into equation (13) yields the following:

$$\mathcal{K}f(x) = \sum_{i=0}^{\infty} \mu_i(-\Delta)^i f = \sum_{i=0}^{\infty} \mu_i \int k_{(-\Delta)^i}(x-y)f(y)\,\mathrm{d}y = \int \left(\sum_{i=0}^{\infty} \mu_i k_{(-\Delta)^i}(x-y)\right) f(y)\,\mathrm{d}y.$$

Comparing the above form of $\mathcal{K}f(x)$ to its original definition $\mathcal{K}f(x) = \int k(x-y)f(y)\,\mathrm{d}y$ implies that we can approximate $k$ as follows

$$
\begin{aligned}
k(x,y) \approx \sum_{i=0}^{\infty} \mu_i k_{(-\Delta)^i}(x-y) &= \sum_{i=0}^{\infty} \mu_i \sum_{j=1}^{\infty} \lambda_j^i \phi_j(x)\phi_j(y) = \sum_{j=1}^{\infty} \left(\sum_{i=0}^{\infty} \mu_i \lambda_j^i\right) \phi_j(x)\phi_j(y) \\
&= \sum_{j=1}^{\infty} s(\sqrt{\lambda_j})\phi_j(x)\phi_j(y),
\end{aligned}
$$

where, in the final step, we utilised our Taylor expansion of the spectral density $s$ of $k$ and set $\|\omega\|^2 = \lambda_j$ for each $j \in \mathbb{N}$. Refer to the original work Solin & Särkkä (2019) for convergence analyses of the given approximation.

Therefore, the resulting Gaussian model assumes the following truncated basis expansion

$$f(\cdot) = \sum_{i=1}^{m} c_i \phi_i(\cdot),$$

where $c_i \sim \mathcal{N}(0, s(\sqrt{\lambda_i}))$ and the $\phi_i$ and $\lambda_i$ are the corresponding eigenfunctions and eigenvalues of the Laplacian over a compact domain $\mathcal{X}$ with Dirichlet boundary conditions $\phi_i(x) = 0$ on $\partial \mathcal{X}$.

When the domain is the unit hypercube, $\mathcal{X} = [0,1]^d$, the resulting eigenfunctions and eigenvalues can be explicitly computed as

$$\phi_j(x) = 2^{d/2} \prod_{k=1}^{d} \sin(\pi j_k x_k), \qquad \lambda_j = \sum_{k=1}^{d} (\pi j_k)^2, \tag{16}$$

where $j = (j_1, \ldots, j_d) \in \mathbb{Z}_m^d$. Taking $m$ sinusoidal functions in each dimension yields $m^d$ eigenfunctions in total. For computational purposes, in `GaussED` the domain $\mathcal{X}$ of the Gaussian model is taken as a $d$-dimensional Cartesian product of intervals $[a_1, b_1] \times \ldots \times [a_d, b_d]$. The required eigenfunctions can be obtained by a simple rescaling of the previous formula.

# E  Computational Details of `GaussED`

In this section we provide details of certain aspects of the computational approaches of `GaussED`. In Appendix E.1, we derive the relevant conditional distributions of the spectral Gaussian process model detailed in Section 2.5, under conditioning on general linear information. In Appendix E.2, we detail the computational approaches of `GaussED` for sampling for the posterior. Finally, in Appendix E.4, we introduce the ANOVA kernel, which is the form of kernel used in the emulation of a Cardiac experiment of Section 3.4.

### E.1   Conditioning

In this section, we both derive and discuss `GaussED`'s approach to conditioning and sampling from the posterior. For completeness, we present the derivation of the conditional distributions of the Gaussian process model detailed in Appendix D. For the sake of generality we consider a general truncated basis model, which takes the form of

$$f(\cdot) = \mu + \sum_{i=1}^{m} c_i \phi_i(\cdot),$$

where the $c_i$ are pairwise independent Gaussian variables and the $\phi_i$ form our basis functions. Suppose that we have a vector of $n$ continuous linear functionals $\boldsymbol{\delta}_n = (\delta_1, \dots, \delta_n)^\top \in \mathcal{D}^n$, such that each $\delta_i$ belong to the design set $\mathcal{D}$ (see Section 2.2). We form the conditional distribution $f \,|\, \boldsymbol{\delta}_n(\mathrm{f})$ as follows, letting $c = (c_1, \dots, c_m)^\top$, we have

$$\begin{pmatrix} c \\ \boldsymbol{\delta}_n(f) \end{pmatrix} \sim \mathcal{N}\left(0, \begin{pmatrix} K_{cc} & K_{c\boldsymbol{\delta}} \\ K_{\boldsymbol{\delta}c} & K_{\boldsymbol{\delta\delta}} \end{pmatrix}\right),$$

where $K_{cc} = \mathbb{C}(c,c) \in \mathbb{R}^{m \times m}$, $K_{c\boldsymbol{\delta}} = \mathbb{C}(c, \boldsymbol{\delta}_n) \in \mathbb{R}^{m \times n}$, $K_{\boldsymbol{\delta}c} = K_{c\boldsymbol{\delta}}^\top$ and $K_{\boldsymbol{\delta\delta}} = \mathbb{C}(\boldsymbol{\delta}_n(f), \boldsymbol{\delta}_n(f)) \in \mathbb{R}^{n \times n}$. The conditional distribution can be computed using standard finite-dimensional formulae as $c \,|\, \boldsymbol{\delta}_n(f) = \boldsymbol{\delta}_n(\mathrm{f}) \sim \mathcal{N}(\mu_{\boldsymbol{\delta}}, \Sigma_{\boldsymbol{\delta}})$, where

$$\mu_{\boldsymbol{\delta}} = K_{c\boldsymbol{\delta}} K_{\boldsymbol{\delta\delta}}^{-1} \boldsymbol{\delta}_n(\mathrm{f}), \tag{17}$$

$$\Sigma_{\boldsymbol{\delta}} = K_{cc} - K_{c\boldsymbol{\delta}} K_{\boldsymbol{\delta\delta}}^{-1} K_{\boldsymbol{\delta}c}. \tag{18}$$

Since the components of $c$ are pairwise independent, we have $K_{cc} = \Lambda = \mathrm{diag}(\mathbb{V}(c_1), \dots, \mathbb{V}(c_m))$. Furthermore, since $\boldsymbol{\delta}_n$ is a vector of linear functionals, we have, for each $i \in \{1, \dots, n\}$, that $\delta_i f = \sum_{j=1}^{m} c_j \delta_i \phi_j$. Therefore, we have

$$\mathbb{C}(\delta_i f, \delta_j f) = \mathbb{C}\left(\sum_{k=1}^{m} c_k \delta_i \phi_k, \sum_{k=1}^{m} c_k \delta_j \phi_k\right) = \sum_{k=1}^{m} \mathbb{V}(c_k) \delta_i \phi_k \delta_j \phi_k$$

and so $K_{\boldsymbol{\delta\delta}} = (\boldsymbol{\delta}\Phi)\Lambda(\boldsymbol{\delta}\Phi)^\top$, where $(\boldsymbol{\delta}\Phi)_{ij} = \delta_i \phi_j$. Finally, we have

$$\mathbb{C}(\delta_i f, c_j) = \mathbb{C}\left(\sum_{k=1}^{m} c_k \delta_i \phi_k, c_j\right) = \mathbb{V}(c_j) \delta_i \phi_j$$

and so $K_{\boldsymbol{\delta}c} = (\boldsymbol{\delta}\Phi)\Lambda$. Thus all the required quantities can be explicitly evaluated.

### E.2   Sampling

To sample from the posterior process $f(\cdot) \,|\, \boldsymbol{\delta}_n(\mathrm{f})$, we can sample from the conditional distribution $c \,|\, \boldsymbol{\delta}_n(\mathrm{f})$ and then utilise the basis expansion of $f$ in (D.2). To achieve this, we are required to perform a matrix square root of the posterior covariance matrix $\Sigma_{\boldsymbol{\delta}}$, and we recall that, when conditioning on exact information, the resulting $\Sigma_{\boldsymbol{\delta}}$ is singular in general. The standard solution of performing a singular value decomposition (SVD) is unsuitable, since the $\Sigma_{\boldsymbol{\delta}}$ often have repeated singular values, which are incompatible with existing implementations of automatic differentiation that assume uniqueness of the singular values (Papadopoulo & Lourakis, 2000; Paszke et al., 2019). Although there have been recent efforts to address this (Wang et al., 2021), the resulting algorithms are computationally prohibitive in our setting.

An alternative method of sampling from $f(\cdot) \,|\, \boldsymbol{\delta}_n(\mathrm{f})$ is called *Matheron's update rule* (Wilson et al., 2021, Corollary 4). Matheron's update rule takes the form

$$f(\cdot) \,|\, \boldsymbol{\delta}_n(\mathrm{f}) \stackrel{d}{=} f(\cdot) + \mathbb{C}(f(\cdot), \boldsymbol{\delta}_n(f)) K_{\boldsymbol{\delta\delta}}^{-1} (\boldsymbol{\delta}_n(\mathrm{f}) - \boldsymbol{\delta}_n(f)). \tag{19}$$

The advantage of Matheron's update rule over the preceding approach is that we are not required to compute the square root of $\Sigma_{\boldsymbol{\delta}}$; this is the default approach used in `GaussED`.

### E.3 Optimising the Acquisition Function

As discussed in Section 2.4, we utilise stochastic optimisation methodology to optimise the acquisition function. Unfortunately, the acquisition functions often exhibit multiple local optima, implying that it is unlikely that the optimiser will find a global optima. There are many approaches to reduce this probability, for instance by running the optimiser at different initialisations in parallel. In `GaussED`, the default approach is to sample uniformly from the design set, then evaluate the acquisition function at each of the sample points, before proceeding to initialise the optimiser at the best obtained point (i.e. Monte Carlo optimisation is used to initialise a stochastic optimisation method). This was the approach used in all the experiments of Section 3.

Since our design sets are based on intervals[8], we perform a standard reparameterisation to obtain a global optimisation problem in $\mathbb{R}^d$. This is achieved by using a scaled logistic function of the form

$$\text{logit}(x; a, b) = \log((x - a)/(b - a)) - \log(1 - (x - a)/(b - a)),$$

where, for $x, a, b \in \mathbb{R}^d$, we consider logit to be applied component-wise. This parametrisation could render it difficult for the optimiser to select design points near the boundaries of the design set; improved methods for stochastic optimisation on bounded domains could be employed within `GaussED` but were not explored in this work.

### E.4 Scalable Kernels

A weakness of the spectral GP approach is that it scales poorly to high-dimensional $\mathcal{X}$, since the number of basis functions required to capture a given frequency scales exponentially in dimension. This occurs only if the kernel has interaction terms between all the input dimensions of the Gaussian process model. Thus, a natural way to scale the Gaussian process model to higher dimensions is to use an ANOVA kernel (Dick & Pillichshammer, 2010), $k : \mathbb{R}^d \times \mathbb{R}^d \to \mathbb{R}$ of the form

$$k(x, y) = \sum_{I \in D} w_I k_I(x_I, y_I), \tag{20}$$

where $D = \bigcup_{i=1}^d \{(j_1, \ldots, j_i) \subseteq \mathbb{N}^i \mid 1 \le j_1 < \ldots < j_i \le d\}$, $w_I \in \mathbb{R}$ are constants, $k_I : \mathbb{R}^{|I|} \times \mathbb{R}^{|I|} \to \mathbb{R}$ is an arbitrary kernel capturing the interactions between the dimensions $I$ and $x_I = (x_{j_1}, \ldots, x_{j_i})$ and $y_I = (y_{j_1}, \ldots, y_{j_i})$. This kernel decomposes all the possible interactions between the input dimensions and is controlled by the constants $c_I$. This allows us to eliminate higher-dimensional interactions to reduce the number of basis functions required. Using our truncated basis expansion and enumerating $D = \{I_1, \ldots, I_N\}$, the ANOVA kernel (20) assumes the latent function takes the form

$$f(x) = \sum_{i=1}^N \sum_{j=1}^{m_i} c_j^i \phi_j^i(x_{I_i}),$$

where the coefficients $c_j^i$ are pairwise independent Gaussian variables. Grouping the coefficients $c = (c^i)_{i=1}^N$, where $c^i = (c_1^i, \ldots, c_{m_i}^i)$, the posterior mean $\mu_\delta$ and covariance matrix $\Sigma_\delta$ of $c \mid \boldsymbol{\delta}_n(f) \sim \mathcal{N}(\mu_\delta, \Sigma_\delta)$ take the same form as (17) and (18) respectively. The constituent matrices $K_{cc}, K_{\delta c}$ and $K_{\delta\delta}$, on the other hand, take the form

$$K_{cc} = \begin{pmatrix} \Lambda^{I_1} & 0 & \ldots & 0 \\ 0 & \Lambda^{I_2} & \ldots & 0 \\ \vdots & \vdots & \ddots & \vdots \\ 0 & 0 & \ldots & \Lambda^{I_N} \end{pmatrix},$$

$$K_{\delta c} = \left( (\delta\Phi^{I_1})\Lambda^{I_1} \quad \ldots \quad (\delta\Phi^{I_N})\Lambda^{I_N} \right), \qquad K_{\delta\delta} = \sum_{i=1}^N (\delta\Phi^{I_i})\Lambda^{I_i}(\delta\Phi^{I_i})^\top,$$

---

[8]Recall from Section 3 that all of the design sets were parameterised as a Cartesian product of intervals.

where $\Lambda^{I_i} = \mathrm{diag}\left(\mathbb{V}(c_1^i), \ldots, \mathbb{V}(c_{m_i}^i)\right)$ and $(\delta^i \Phi^{I_i})_{jk} = \delta_j \phi_k^i$, where $\delta_j^i$ is the functional $\delta_j$ restricted to the input dimensions $I_i$.

An ANOVA kernel was used in the emulation of a Cardiac model experiment in Section 3.4 and detailed in Appendix F.4.

## F  Experimental Details

In this section we present full details for the experiments presented in Section 3. Appendix F.1 details the PDE experiment presented in Section 3.1. Appendix F.2 details the tomographic reconstruction experiment presented in Section 3.2. Appendix F.3 details the Lotka–Volterra experiment presented in Section 3.3. Appendix F.4 details the emulation of the Cardiac model presented in Section 3.4.

### F.1  Probabilistic Solution of PDEs

**Approximating the Loss:**  Following from Section 3.1, recall that the quantity of interest was the function f, implying the loss takes the form

$$L(g, g') = \|g - g'\|^2 = \int_{\mathcal{X}} |g(x) - g'(x)|^2 \, \mathrm{d}x.$$

Since there is not a closed-form solution to this integral when $g$ is a Gaussian process, we proceed by approximating the integral through a cubature rule. For this experiment, we performed a Riemann sum over a uniform $15 \times 15$ grid over the domain $\mathcal{X} = [-1, 1]^2$.

**Gaussian Model:**  For this experiment, we used a mean-zero Gaussian process $f$ with Matérn covariance function with smoothness parameter $\nu = 3$. The Dirichlet boundary conditions of the PDE were automatically enforced by the spectral GP approximation, applied to the domain $\mathcal{X} = [-1, 1]^2$ (c.f. Equation (15)).

**Optimisation:**  For both the optimisation of the acquisition function and performing maximum likelihood estimation, we used the Adam stochastic optimisation methodology (Kingma & Ba, 2015).

Using the methodology discussed in Appendix E.3, at each iteration of SED, we sampled 100 points uniformly from the design set and computed the corresponding values of acquisition function, using the default values of $N = 81$ and $M = 9$ in the stochastic gradient estimator of Section 2.4. We then proceeded by initialising the stochastic optimiser at the sample point which minimised the acquisition function. The learning rate used was the default value of $10^{-1}$ and the optimiser was run for 1000 iterations, at each step of SED. The SED began with an initial design consisted of evaluations of the Laplacian over a grid of 9 evenly spaced points over the domain $[-0.98, 0.98] \times [-0.98, 0.98]$.

Using the methodology as discussed in Section 2.6, we began optimising the amplitude $\lambda$ and the lengthscale $\ell$ after $n_0 = 10$ iterations of SED. This $n_0 = 10$ is the default value in `GaussED`. The initial parameter values were taken as the default values of $\lambda = \exp(1/2)$ and $\ell = 0.2$. The learning rate used was the default value of $10^{-3}$ and the optimiser was run for 1000 iterations, at each step of SED.

**Code:**  The code used to run the experiment can be seen in Figure 1 and discussed in Section 3.1.

### F.2  Tomographic Reconstruction

**Approximating the Loss:**  Following from Section 3.2, recall that the quantity of interest was the function $\exp(3f)$, implying the loss takes the form

$$L(g, g') = \|\exp(3g) - \exp(3g')\|^2 = \int_{\mathcal{X}} |\exp(3g(x)) - \exp(3g'(x))|^2 \, \mathrm{d}x.$$

We follow the same approach of Appendix F.1 and approximate the integral through a Riemann sum, now over a uniform $25 \times 25$ grid over the domain $\mathcal{X} = [-1, 1]^2$.

```
k = MaternKernel(2, 2, initial_parameters)
domain = [[-1.05,1.05],[-1.05,1.05]]
gp = SpectralGP(k)
gp.set_domain(domain)

exponential_warp = lambda x: torch.exp(3 * x)
qoi = OutputWarp(gp, exponential_warp)()

X,Y = torch.meshgrid(torch.linspace(-1,1,25),torch.linspace(-1,1,25))
mesh = torch.stack([X,Y]).T.reshape(25**2,2)

loss = L2(qoi, mesh)

def d_func(design, m):
    all_phis = []
    for i in range(len(design)):
        design_i = design[i]
        line_int_gps = get_line_int_gps(design_i.unsqueeze(1), gp)
        for j in line_int_gps:
            all_phis.append(j.basis_matrix(None,m))
    return torch.cat(all_phis)

def d_sample(design_point, mean, cov, n, random_sample=None):
    all_samples = []
    line_int_gps = get_line_int_gps(design_point.unsqueeze(1), gp)
    matrix_sqrt = gp.solver.square_root(cov)
    for i in line_int_gps:
        samp_i = i.sample(mean, cov, n, random_sample, sqrt=matrix_sqrt)(None)
        all_samples.append(samp_i)
    return torch.cat(all_samples).T

initial_design = torch.Tensor([[0,  0,  0]])
d = Design(d_func, d_sampling, initial_design)
d.set_domain([[0, math.pi],[-1,1],[-1,1]])

acq = BayesRisk(qoi, loss, d, nugget=1e-2)
experiment = Experiment(gp, transformed_black_box, d, acq, m=28)
experiment.run(30)
```

Figure S1: The `GaussED` code used to run the tomographic reconstruction experiment of Section 3.2.

**Gaussian Model:**  For this experiment, we utilised a stationary Gaussian process model $f$ with Matérn covariance with smoothness parameter $\nu = 2$. The Gaussian model is defined on the domain $[-1.05, 1.05]^2$, since the boundary conditions of the resulting GP do not necessarily agree with the boundary conditions of the quantity of interest.

**Quantity of Interest:**  Recall from Section 3.2 that the quantity of interest was of the form

$$\mathsf{f}(x) = \begin{cases} 1, & \text{when } \|x - (0.4, 0.4)\| < 0.3, \\ 0, & \text{otherwise.} \end{cases}$$

Since this quantity of interest defines a circle within the domain $\mathcal{X} = [-1, 1]^2$, it is possible to find a closed form solution to the line integrals of $\mathsf{f}$ for given parameters values $(\theta, x, y)$. However, for ease of

implementation and to allow our approach to be easily generalised to more complex examples, we computed the line integrals of f by performing a Riemann integral over a uniform mesh consisting of 200 evaluations from f.

**Optimisation:**    For both the optimisation of the acquisition function and performing maximum likelihood estimation, we used the Adam stochastic optimisation methodology (Kingma & Ba, 2015).

Using the methodology discussed in Appendix E.3, at each iteration of SED, we sampled 400 points uniformly from the design set and computed the corresponding values of acquisition function, using the default values of $N = 81$ and $M = 9$ in the stochastic gradient estimator of Section 2.4. We then proceeded by initialising the stochastic optimiser at the sample point which minimised the acquisition function. The learning rate used was the default value of $10^{-1}$ and the optimiser was run for 1000 iterations, at each step of SED. The SED began with an initial design consisted of 9 line integral evaluations over the vertical lines $x = -0.09, -0.03, 0, 0.03, 0.06, 0.09$.

Using the methodology as discussed in Section 2.6, we began optimising the amplitude $\lambda$ and the lengthscale $\ell$ after $n_0 = 5$ iterations of SED. The initial parameter values were taken as the default values of $\lambda = \exp(0.3)$ and $\ell = 0.4$. The learning rate used was the default value of $10^{-3}$ and the optimiser was run for 1000 iterations, at each step of SED.

**Code:**    The `GaussED` code used to run this experiment is presented in Figure S1. The structure of the code is quite different to the code used in the other experiments (Figure 1 and Figure S3). This is due to the fact that the design object (`d`) is not instantiated by the `EvaluationDesign` class. Note that for both the PDE experiment (Section 3.1) and the Bayesian optimisation experiment (Section 3.3), the design sets $\mathcal{D}$ consisted of evaluations of the Gaussian process, $\delta(f) = f(x)$, or its derivatives $\delta(f) = \partial_i f(x)$. In situations such as these, the `EvaluationDesign` class may be used. For this example, however, the observed data consists of line integrals. Therefore, in this more general situation, we must specify two further functions: Given a parameterisation $\mathcal{D}_\theta$ of the design set, the first function must take in a sequence of parameters $\theta_1, \ldots, \theta_n$ and return the corresponding $(\boldsymbol{\delta\Phi})_{ij} = \delta_{\theta_i}\phi_j$ matrix, where the $\phi_j$ are the eigenfunctions of (16). This is reflected in the code (Figure S1) in the function `d_func`, which, for each design set parameter constructs the corresponding line integral for a given number of basis functions (`m`). The second function we must specify must be able to, given a parameter $\theta$, sample from the process $\delta_\theta f \,|\, \boldsymbol{\delta}_n$, where $\boldsymbol{\delta}_n$ is data gathered from SED. This is directly reflected in the code (Figure S1) in the function `d_sample`. Note that, in Figure S1 we omit the `get_line_int_gps` function. This is a function that, given a parameter value $\theta$ and Gaussian process $f$, returns the corresponding $\delta_\theta f$ object. We do this because `get_line_int_gps` is complexified due to the parameterisation of the line function $r(x)$ and the calculation of the limits of integration $a, b$ in the line integral

$$\int_a^b f(r(x)) \,\mathrm{d}x.$$

We, therefore, omit `get_line_int_gps` for clarity.

A second major difference, is the use of an output warp (`OutputWarp`). Due to the non-linear nature of the output warp, the resulting object `qoi` is only able to sample from the prior and posterior. Note that the syntax for specifying a output deformation of a GP is the same as specifying other transformations (e.g. see Figure 1 and Figure S3).

Another difference is that the Gaussian model specified in the PDE experimental code (Figure 1) agreed with the boundary conditions of the PDE; here, however, we specify the domain (`gp.set_domain`) as $[-1.05, 1.05] \times [-1.05, 1.05]$. Since we took the domain of the Gaussian process to be larger than the domain on which the task is defined, we must also specify the domain of the design object (`d.set_domain`), which otherwise, by default, would be taken as the same the Gaussian model (`gp`).

Finally, note that the acquisition function (`acq`), as discussed previously, is instantiated with a nugget value of $10^{-2}$ and the experiment object (`experiment`) is instantiated with $m = 28^2$ basis functions. This is in contrast to the code for the PDE example (Figure 1), which used the default value of $m = 30^2$ basis functions.

### F.3 Gradient-Based Bayesian Optimisation

**Approximating the Loss:**   Recall from Section 3.3 that our quantity of interest is $q(\mathsf{f}) = \max_{x \in \mathcal{X}} \log \mathcal{L}(x)$. Thus, our loss function takes the form

$$L(g, g') = \left| \max_{x \in \mathcal{X}} (g(x)) - \max_{x \in \mathcal{X}} (g'(x)) \right|^2 .$$

In order to optimise the samples, we used a grid-based optimiser using a uniform $40 \times 40$ grid over the domain of interest $[0.45, 0.9] \times [0.09, 0.5]$.

**Gaussian Model:**   For this experiment, we used a mean-zero stationary Gaussian model $f$ with Matérn covariance, with smoothness parameter $\nu = 3$. Since our GP satisfies the boundary conditions in (15), which are unrelated to the task at hand, we took the domain of the GP to be $[0.4, 0.95] \times [0.04, 0.55]$, which is wider than the domain on which the task is defined.

**Quantity of Interest:**   Synthetic data $y = (p_i, q_i)_{i=1}^{51}$ were generated at times $t = 0, 0.5, 1, \ldots, 50$ by perturbing the solution of the Lotka–Volterra model, with parameter values $(\alpha, \beta, \gamma, \delta) = (0.5, 0.1, 0.3, 0.1)$, with mean-zero Gaussian errors with variance $\sigma^2 = 0.05^2$. The data used for the log-likelihood and the corresponding true solution with $(\alpha, \beta, \gamma, \delta) = (0.5, 0.1, 0.3, 0.1)$ are displayed in Figure S2.

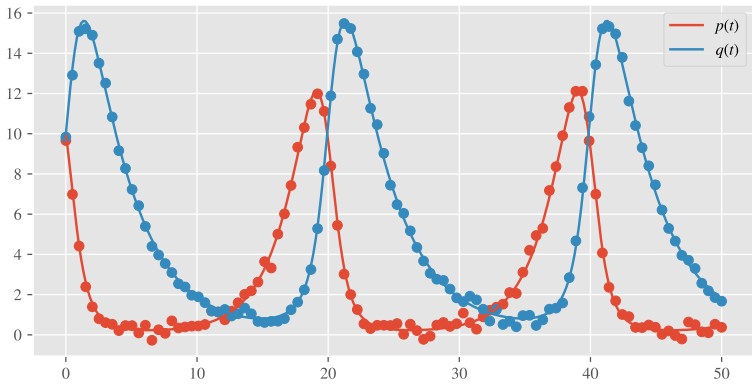

Figure S2: Solution of the Lotka–Volterra model with parameter values $\theta = (0.5, 0.1, 0.3, 0.1)$, with the synthetic data $y = (p_i, q_i)_{i=1}^{51}$ overlaid.

**Optimisation:**   For both the optimisation of the acquisition function and performing maximum likelihood estimation, we used the Adam stochastic optimisation methodology (Kingma & Ba, 2015).

Using the methodology as discussed in Appendix E.3, at each iteration of SED, we sampled 100 points times uniformly from the design set and computed the corresponding values of acquisition function, using the default values of $N = 81$ and $M = 9$ in the stochastic gradient estimator of Section 2.4. We then proceeded by initialising the stochastic optimiser at the sample point which minimised the acquisition function. The learning rate used was the default value of $10^{-1}$ and the optimiser was run for 1000 iterations, at each step of SED. The SED began with an initial design consisting of the evaluation of the log-likelihood $\mathcal{L}$ and its gradient at the midpoint of the domain $(\alpha, \beta) = (0.675, 0.295)$. Finally, in order to increase the numerical stability of linear algebra operations, we used a nugget term of value $10^{-5}$.

For this experiment, we began optimising the amplitude $\lambda$ and lengthscale $\ell$ at step $n_0 = 10$. The initial kernel parameter values were taken as the values of $\lambda = 1$ and $\ell = 0.1$. The learning rate used was the default value of $10^{-3}$ and the optimiser was run for 1000 iterations, at each step of SED.

**Code:**   The `GaussED` code used to run this experiment is presented in Figure S3. The structure of the program is very similar in nature to the PDE experiment of Section 3.1. The first difference is that, at each

step of SED, we evaluate multiple functionals $\delta$ from the design $\mathcal{D}$. This is directly reflected in Figure S3, where the design object (`d`) is constructed by the statistical model $f$ and its first derivatives (`[gp, gp_d1, gp_d2]`).

The second difference is that, at each step SED, we perform a maximisation, rather than an integral, of sample paths when estimating the acquisition function. In the code for the PDE experiment (Figure 1) the integral of posterior samples is hidden within the loss object (`L2(qoi)`), which, by default, performs a Riemann sum over a uniform mesh if the quantity of interest `qoi` is function valued. Therefore, in Figure S3 we specify a numerical method that acts on samples from $f$. In this instance, we perform a grid search (`maximise_method`) over a uniform $40 \times 40$ mesh (`mesh`) over the domain of optimisation.

Another difference is that the Gaussian model specified in the PDE experimental code (Figure 1), agrees with the boundary conditions of the PDE and therefore the domain of the GP is taken as the default value $[-1, 1]^2$. In Figure S3, we must specify the domain (`gp.set_domain`) as $[0.4, 0.95] \times [0.04, 0.55]$. Since we took the domain of the Gaussian process to be larger than the domain over which we wish to maximise, we must also specify the domain of the design object (`d.set_domain`), which otherwise, by default, would be taken as the same the Gaussian model (`gp`).

The final difference is that, in order to increase the numeric stability of linear algebra operations in the SED, we specify a nugget term (`nugget`) of value $10^{-5}$ in the acquisition function `acq`.

```
k = MaternKernel(3, 2, initial_parameters)
gp = SpectralGP(k)
gp.set_domain(torch.Tensor([[0.4,0.95],[0.04,0.55]]))

gp_d1 = Differentiate(gp,[0],[1])()
gp_d2 = Differentiate(gp,[1],[1])()

x, y = torch.meshgrid(torch.linspace(0.45,0.9,40),torch.linspace(0.09,0.5,40))
mesh = torch.stack([x,y]).T.reshape(40**2,2)
maximise_method = GridSearch(mesh)

qoi = Maximise(gp, maximise_method)()

d = EvaluationDesign([gp, gp_d1, gp_d2], init_design)
d.set_domain(torch.Tensor([[0.45,.9],[0.09,0.5]]))

loss = L2(qoi)
acq = BayesRisk(qoi, loss, d, nugget=1e-5)

experiment = Experiment(gp, lotka_volterra, d, acq, m=35)
experiment.start_hyp_optimising_step = 10
experiment.run(30)
```

Figure S3: The `GaussED` code used to run the gradient-based Bayesian optimisation experiment of Section 3.3.

### F.4 Emulation of a Cardiac Model

**Approximating the Loss:** Recall from Section 3.4 that our quantity of interest is $q(f) = f$. Thus, our loss function takes the form

$$L(g, g') = \|g - g'\|^2 = \int_{\mathcal{X}} |g(x) - g'(x)|^2 \, \mathrm{d}x,$$

We follow the same approach as Appendix F.1 and Appendix F.2 and approximate this integral through a Riemann sum, now over a uniform $5 \times 5 \times 5 \times 5$ grid over the domain $\mathcal{X} = [0.1, 5]^4$. It was pointed out by

```python
anova_kernel = AnovaKernel(MaternKernel(2.5, 4))
m_list = [25,25,25,25,7,7,7,7,7,7,2,2,2,2,2]

l_scales = torch.Tensor([[0.5],[0.5],[0.5],[0.5]]).requires_grad_(True)

gp = SpectralGP(anova_kernel)
gp.set_domain([[0,5.1] for i in range(4)])

def update_gp_params():
    gp.kernel.parameters[1::2] = [l_scales[i].norm() for i in interacting_dims]

update_gp_params()

noise_variance = torch.Tensor([1]).requires_grad_(True)
all_params = [l_scales] + gp.kernel.parameters[0::2] + [noise_variance]

qoi = Maximise(gp, maximise_method)()

x = torch.linspace(0.1,5,5)
X1,X2,X3,X4 = torch.meshgrid(x,x,x,x)
mesh = torch.stack([X1,X2,X3,X4]).T.reshape(5**4,4)
loss = L2(gp, mesh)

d = EvaluationDesign(gp, init_design)
acq = BayesRisk(q, loss, d, nugget=noise_variance)

def hyper_reg(alpha, beta):
    l_scales_reg = alpha
            * (1 / torch.cat([l_scale[i].pow(2) for i in interacting_dims])).max()
    amplitude_reg = beta * torch.cat(gp.kernel.parameters[0::2]).exp().norm()
    return l_scales_reg + amplitude_reg

def parameter_objective(a, b, alpha=0.0001, beta=5):
    return - gp.log_likelihood(a, b, nugget=noise_variance, m=m_list)
            + hyper_reg(alpha,beta)

hyper_optim = DefaultOptimiser(parameter_objective, torch.optim.Adam, all_params)
hyper_optim.set_optional_func(update_gp_params)

experiment = Experiment(gp, black_box, d, acq, m=m_list, hyper_optim=hyper_optim)
experiment.acq_optim.N = 200
experiment.acq_optim_steps = 600
experiment.hyper_optim_steps = 100
experiment.run(100, optimise_hyper=False)
experiment.run(900)
```

Figure S4: The `GaussED` code used to run the emulation of a Cardiac model experiment of Section 3.4.

one reviewer that a relatively coarse set of cubature nodes on which the Riemann sum is calculated could lead to the accumulation of experimental design points around the cubature nodes; this can in principle be avoided by using a more accurate cubature rule, but for the present paper this was not pursued.

**Gaussian model:**  For this experiment we utilise an ANOVA kernel as described in Appendix E.4. Each sub-kernel $k_I$ for $I \in D$ were parameterised with an amplitude parameter $\lambda_I$ and lengthscale parameter $\ell_I$ that was further parameterised as a function of four characteristic lengthscales $\ell_1, \ell_2, \ell_3$ and $\ell_4$, one for each dimension. We defined $\ell_I$ as follows:

$$\ell_I := \sqrt{\sum_{i \in I} (\ell_i)^2}.$$

Each of the sub-kernels $k_I$ were taken as the Matérn covariance function with smoothness parameter $\nu = 2.5$ and the domain of the GP was taken as $[0, 5.1]^4$.

**Optimisation:**  For both the optimisation of the acquisition function and performing (regularised) maximum likelihood estimation, we used the Adam stochastic optimisation methodology (Kingma & Ba, 2015).

Using the methodology as discussed in Appendix E.3, at each iteration of SED, we sampled 200 points times uniformly from the design set and computed the corresponding values of acquisition function, using the default values of $N = 81$ and $M = 9$ in the stochastic gradient estimator of Section 2.4. We then proceeded by initialising the stochastic optimiser at the sample point which minimised the acquisition function. The learning rate used was the default value of $10^{-1}$ and the optimiser was run for 600 iterations, at each step of SED. The SED began with an initial design consisting of the evaluation of the cardiac model at the midpoint of the domain $x = (2.55, 2.55, 2.55, 2.55)$. Due to the noisy observations, we also introduced a nugget term $\sigma^2$, with initial value $\sigma^2 = 1$. The nugget term was incorporated as a hyperparameter and optimised using (regularised) maximum likelihood estimation, described next:

To prevent overfitting for small data, we introduced a regulariser term in the maximum likelihood estimation of the hyperparamters, taking the form

$$r_{\alpha, \beta}(\boldsymbol{\lambda}, \boldsymbol{\ell}) = \alpha \max_i |1/\ell_i| + \beta \|\boldsymbol{\lambda}\|,$$

where $\boldsymbol{\lambda} = (\lambda_I)_{I \in D}$ and $\boldsymbol{\ell} = (\ell_1, \dots, \ell_4)$ are collection of amplitude and characteristic lengthscale parameters respectively. Such a regulariser, when minimised, prevents the lengthscales growing too small and amplitudes growing too large. For this experiment we took $\alpha = 10^{-4}$ and $\beta = 5$ and begun hyperparameter optimisation of all hyperparmaters at step $n_0 = 100$. The initial parameter values were taken as the values of $\lambda_I = 1$ for $I \in D$ and $\ell_1 = \ell_2 = \ell_3 = \ell_4 = \exp(0.3)$. The learning rate used was the default value of $10^{-3}$ and the optimiser was run for 100 iterations, at each step of SED.

**Code:**  The `GaussED` code used to run this experiment is presented in Figure S4. The structure of the program is similar in nature to the previous examples. The key differences in this case are the following:

Firstly, due to lengthscales of each of the sub-kernels $k_I$ being further parameterised by characteristic lengthscales $\boldsymbol{\ell}$, we introduce the function `update_gp_params` which updates the lengthscales of each of the kernel $k_I$. Here `interacting_dims` is a list corresponding to the set $D$ interacting dimensions $I$. For legibility, the value of the variable is not presented. This function is called at each gradient update of the characteristic lengthscales $\boldsymbol{\ell}$ in order to update the corresponding lengthscales of the sub-kernels $k_I$ and thus is set as an `optional_func` in `hyper_optim`.

Secondly, since the hyperparameters are optimised using a regularised maximum-likelihood, we must define the hyperparameter optimiser (`hyper_optim`), which in the other experiments by default performs maximum-likelihood estimation. The regulariser term corresponds to the function `hyper_reg`.

Finally, further experiment options are used, where the number of steps taken at each iteration of SED when optimising the acquisition function is taken as 600 (`acq_optim_steps`) and the number of steps taken at each iteration of SED when optimising the (regularised) log-likelihood function is taken as 100 (`hyper_optim_steps`).

# G Evaluating Computational Aspects of `GaussED`

In this section we empirically investigate computational aspects of `GaussED`. In Appendix G.1, we explore the role of the optimisation methodology and how this affects the experimental design as well as the quality of output. In Appendix G.2, we investigate how the number of basis functions used, for a given problem, affects the quality of posterior inference. In Appendix G.3, we compare the acquisition functions expected improvement and Bayes risk in the context of Bayesian optimisation.

## G.1 Investigating the Efficacy of Stochastic Optimisation

In this section, we investigate the effect of the random seed on the quality of the experimental design and, further, investigate the effect of changing the stochastic optimisation approach itself. To explore these aspects of `GaussED`, we repeat the Bayesian optimisation with gradient data experiment presented in Section 3.3. Recall that, in all the demonstrations in Section 3, we utilised the Adam stochastic optimisation method (Kingma & Ba, 2015).

Results on the effect of the random seed can be seen in Figure S5 and Figure S6. The obtained designs imply that our approach of SED is sensitive to the initial conditions. Although the specific design is sensitive, the overall performance and qualitative nature of the designs are approximately independent of random seed.

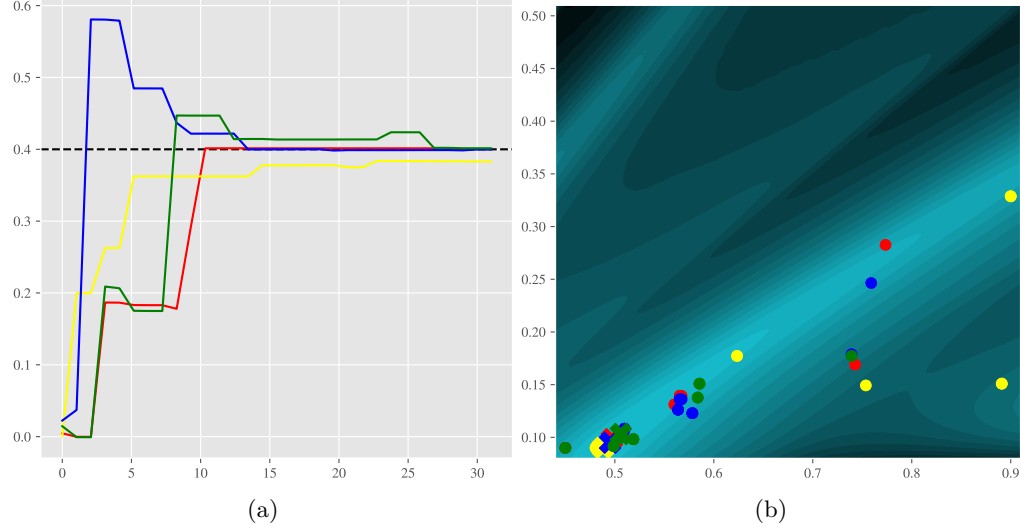

(a) (b)

Figure S5: Convergence analysis of Bayesian optimisation with 4 different random seeds. The left panel (a) displays the maximal value obtained for each of the random seeds, with each colour corresponding to a different random seed. The right panel (b) displays the coordinate positions of the obtained maximum value, where the colored symbols ✖ indicate the coordinate position of the obtained maximum value at termination. Again, the maximum of the posterior mean is reported.

Results on the effect of stochastic optimisation methodology can be seen in Figure S7 and Figure S8. In each of these experiments, the random seed was fixed, and so we are only comparing the effect of different optimisation methodologies. In each experiment, the learning rate was set at $10^{-1}$ and the other parameter values were taken as their default values, as specified in `PyTorch` (Paszke et al., 2019).

## G.2 Investigating the Effect of the Number of Basis Functions

Picking an appropriate number of basis functions for a given problem is an important means to reduce computational cost in `GaussED`. In this section, we investigate how the number of basis functions may affect the quality of posterior inference. To this end, it is sufficient to consider the behaviour of posterior sampling in dimension $d = 1$, since the behaviour will naturally extend to higher-dimensions due to the exponential scaling of the number of basis function due to (16).

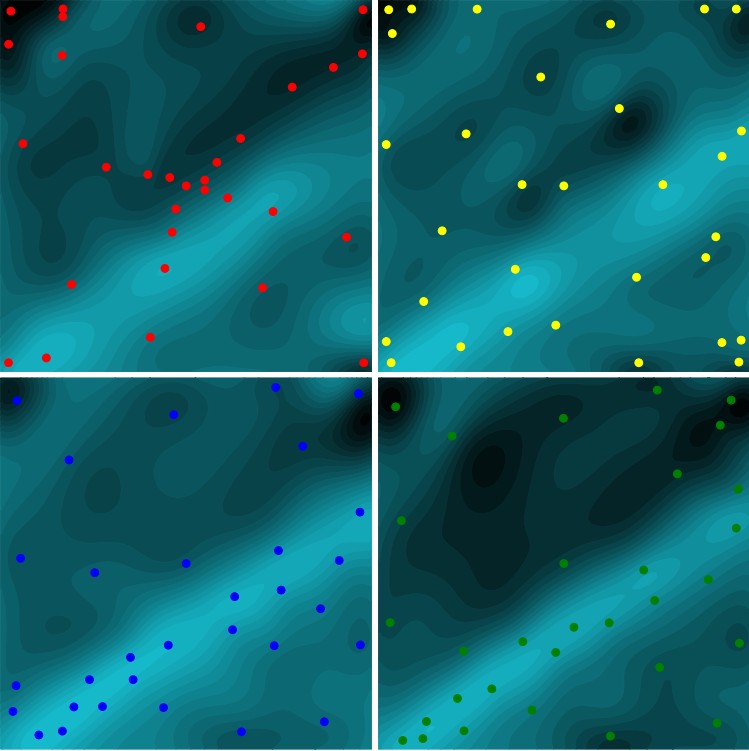

Figure S6: Designs obtained by SED for the 4 different random seeds along with the corresponding obtained posterior means. The colours correspond to the same random seed as displayed in Figure S5.

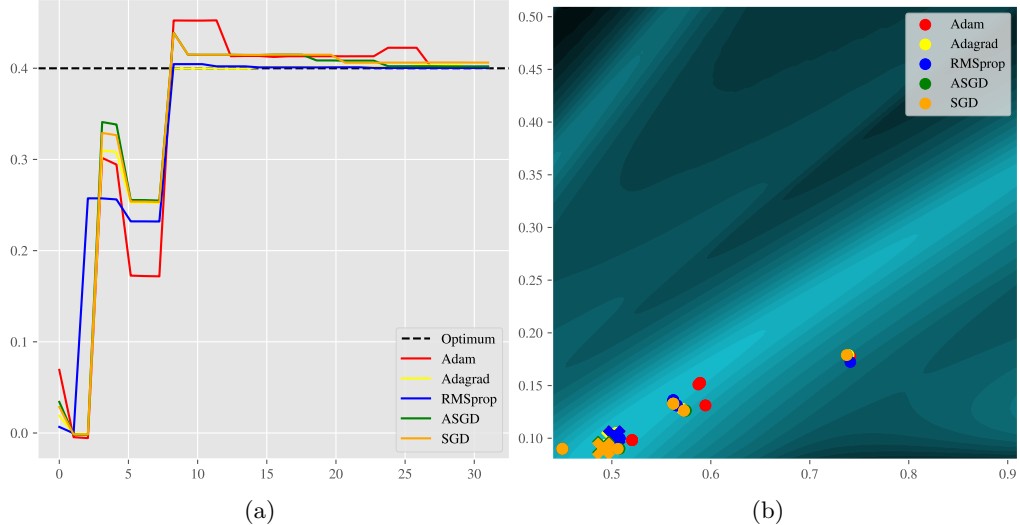

(a)                                           (b)

Figure S7: Convergence analysis of Bayesian optimisation with different optimisation methods. The left panel (a) displays the maximal value obtained for each of the optimisation methods, with each colour corresponding to a different method. The right panel (b) displays the coordinate positions of the obtained maximum value, where the colored symbols ✖ indicate the coordinate position of the obtained maximum value at termination. Again, the maximum of the posterior mean is reported.

In the event where the number of basis functions is smaller than the number of linearly independent data, the resulting posterior will not be well-defined in general. The introduction of a nugget term on the diagonal of the covariance matrix, implicitly assuming noisy Gaussian observations, is a pragmatic solution that is

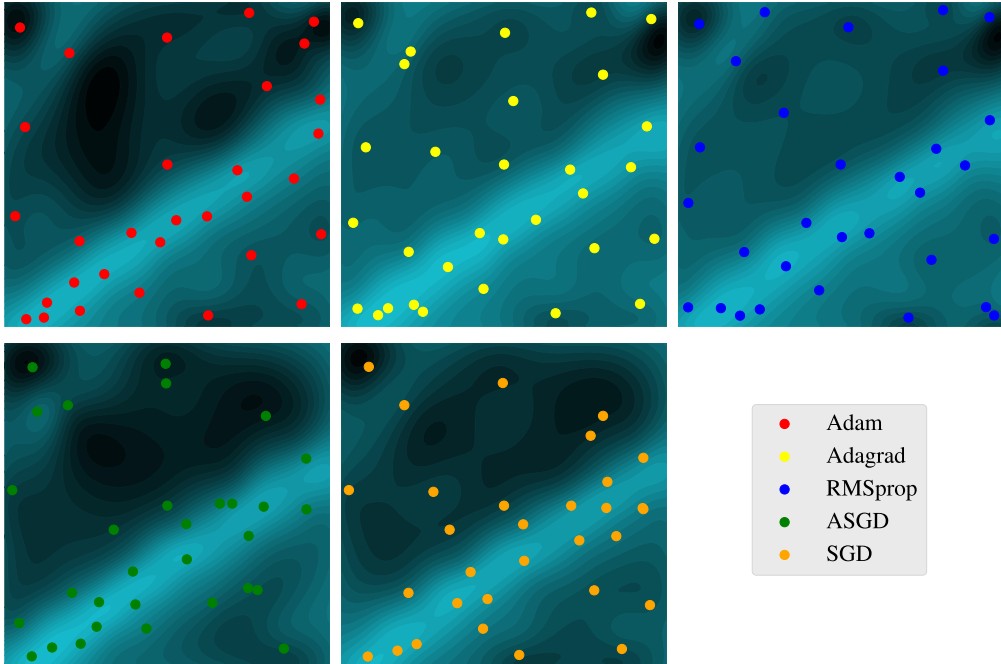

Figure S8: Designs obtained by SED for the 5 different optimisation methods along with the corresponding obtained posterior means. The colours correspond to the same optimisation methods as displayed in Figure S7.

widely-used. However, the success of this strategy depends crucially on using an appropriate amount of regularisation.

Results on the effect on the number of basis functions and the nugget term are presented in Figure S9. Through visual inspection, by $m = 20$ basis functions, it appears that the posterior process has converged sufficiently well to the true posterior process. Note that, when $m = 7$, the posterior sample paths overlap. This is due to there being only one value of $c_1, \ldots, c_7$ such that the truncated basis model agrees with the 7 evaluations.

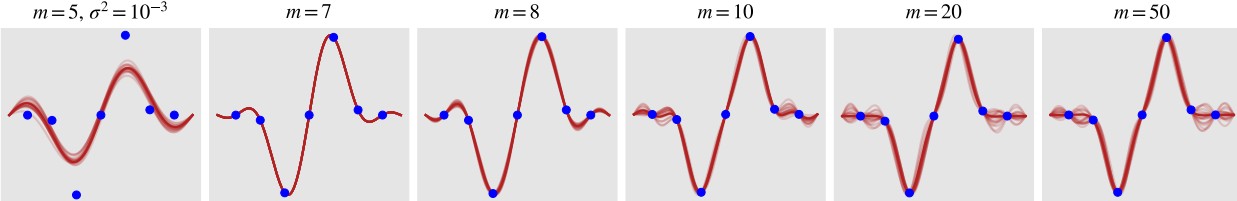

Figure S9: Samples and posterior mean based on a mean-zero Gaussian process $f$ with Matérn covariance with smoothness parameter $\nu = 1.5$, amplitude $\lambda = 0.1$ and lengthscale $\ell = 0.1$, conditioned to interpolate the 7 (blue) data points. The corresponding number $m$ of basis functions used in each experiment is displayed in the titles of the subplots. In the event where $m$ is smaller than the number of data points conditioned upon, the corresponding nugget term $\sigma^2$ is also displayed.

## G.3 Comparison with Expected Improvement

In this section we compare we compare Bayesian optimisation using the Expected Improvement (EI) acquisition function (Jones et al., 1998) against Bayes risk (see Section 2.3) with the non-linear quantity of interest $\max_{x \in \mathcal{X}}(f(x))$. The standard setting of Bayesian optimisation is setting $f \sim \mathcal{GP}(m, k)$ and conditioning on evaluation functionals $\delta f = f(x)$ to obtain the maximum value of a latent function $f$. Given $n - 1$ evaluation

functionals $\boldsymbol{\delta}_{n-1}(\mathsf{f}) = (\mathsf{f}(x_1), \ldots, \mathsf{f}(x_n))$, the expected improvement acquisition function, EI, is defined as

$$\mathrm{EI}(x; \mathbb{P}_f, \boldsymbol{\delta}_{n-1}(\mathsf{f})) = \mathbb{E}\left[\max(f(x) - \max_i \mathsf{f}(x_i), 0) \,|\, \boldsymbol{\delta}_{n-1}(\mathsf{f})\right].$$

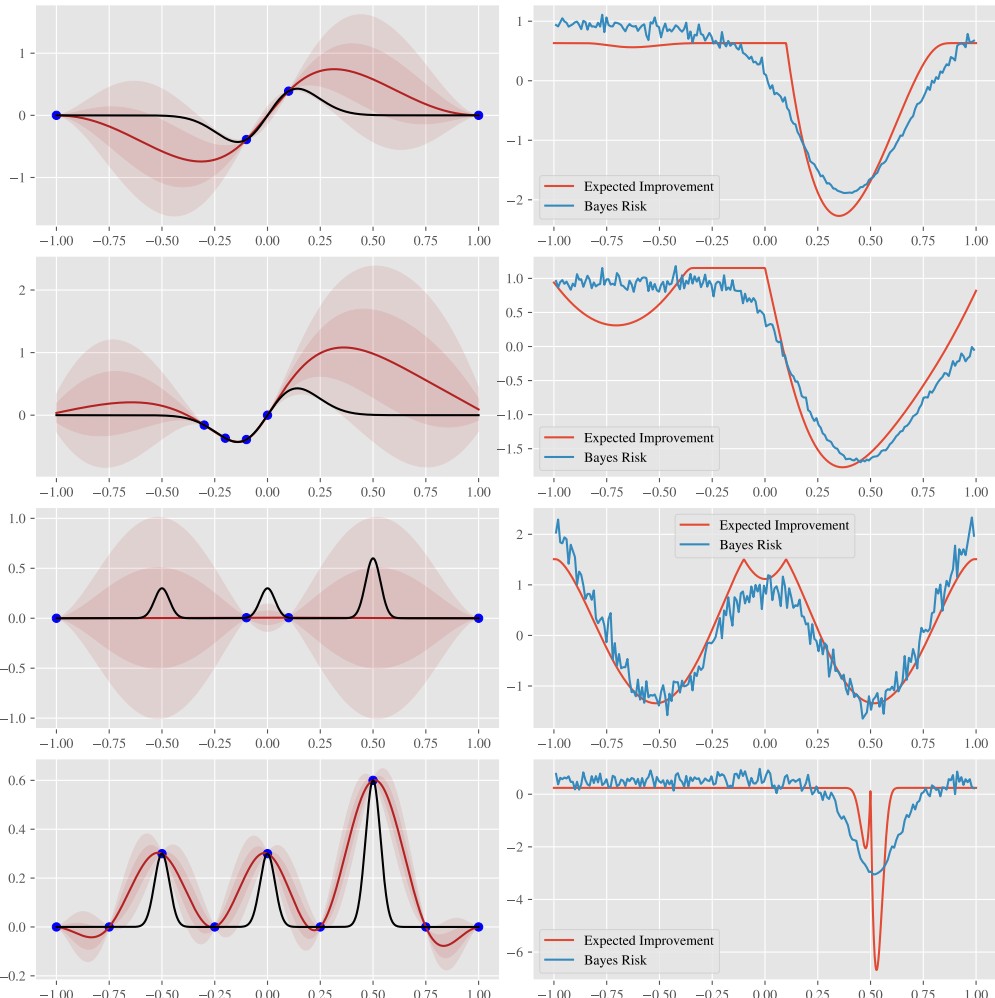

Figure S10: *Comparing* EI *against Bayes risk:* Left: Plot of the latent function (black) against the posterior mean (red) with pointwise credible intervals indicated in shaded red. Right: Plot of the corresponding normalised expected improvement (red) and normalised Bayes risk (blue). The first two rows were obtained using evaluations from $\mathsf{f}_1$ (21) and the subsequent rows were obtained using evaluations from $\mathsf{f}_2$ (22).

In order to investigate the practical differences between EI and Bayes risk, we plot the acquisition functions evaluated over the design space $x \in \mathcal{X}$ in a few scenarios of two different one-dimensional functions. We took

$$\mathsf{f}_1(x) = 5x \exp(-(5x)^2), \tag{21}$$

$$\mathsf{f}_2(x) = 0.3\left(\exp\left\{-(20(x+0.5))^2\right\} + \exp\left(-(20x)^2\right\} + 2\exp\left(-(20(x-0.5))^2\right\}\right). \tag{22}$$

We used a mean-zero Gaussian process with Matérn covariance with smoothness parameter $\nu = 2.5$. The Gaussian process was defined over the domain $[-1.05, 1.05]$ and we specified its amplitude as $\lambda = 1$ and lengthscale as $\ell = 0.4$. Since the Bayes risk lacks a closed form, we evaluated it with parameters $N = 900$ and $M = 30$ in the following Monte-Carlo approximation

$$A(\delta; \mathbb{P}_f, \boldsymbol{\delta}_{n-1}(\mathsf{f})) \approx -\frac{1}{2}\frac{1}{NM}\sum_{i=1}^{N}\sum_{j=1}^{M} L(g_i, \eta(\omega_{ij}, \mathbb{P}_f, \boldsymbol{\delta}_{n-1}(\mathsf{f}), \delta_z(g_i))),$$

where the $g_i$ and $\eta$ are defined as in (7). The loss function was approximated by maximising the GP samples using grid-based optimiser over a uniform length 200 mesh over the domain $[-0.99, 0.99]$. Results are reported in Figure S10. Note that, unlike the Bayes risk, expected improvement tends to be flat over subregions of the domain, indicating expected improvement is less suitable for gradient based optimisation methods. Due to this, in the subsequent experiment, we first evaluate expected improvement over a randomly generated set of $N = 100$ points, before proceeding to gradient based optimisation.

We further investigate performing optimisation on the two acquisition functions to compare their practical performance in a SED scenario. We considered the optimisation of a two-dimensional mixture of Gaussians, each centered over a $3 \times 3$ uniform grid over the domain $[-0.75, 0.75] \times [-0.75, 0.75]$. We used a mean-zero Gaussian process with Matérn covariance with smoothness parameter $\nu = 2.5$. The Gaussian process was defined over the domain $[-1.05, 1.05] \times [-1.05, 1.05]$ and we specified its amplitude as $\lambda = 1$ and lengthscale as $\ell = 0.2$. We ran each experiment for 25 iterations of SED, using the standard settings as described in Appendix F. Results are shown in Figure S11. It should be noted that expected improvement tends to behave more exploitatively, whereas Bayes risk tends to favour exploration.

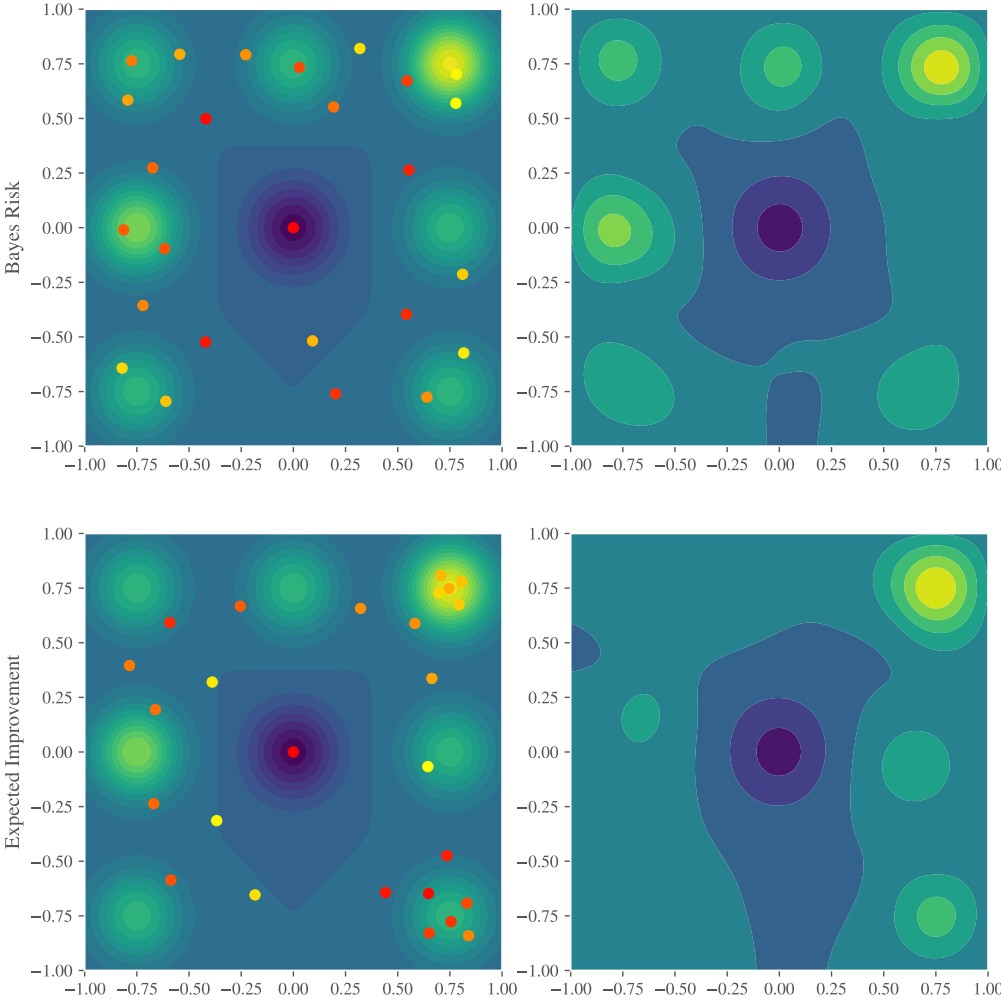

Figure S11: *Comparing* EI *against Bayes risk:* Left hand displays a contour plot of the latent function f with the design points, either determined through Bayes risk or EI, overlaid. Right hand displays the corresponding posterior mean plots.

