# OpenReview forum: "GaussED:  A Python Package for Sequential Experimental Design"
_TMLR — Rejected by TMLR_

### Review · Reviewer_R11h · 2024-03-11

**Summary Of Contributions:**

The paper introduces a framework to automate Sequential Experimental Design (SED) for Gaussian processes in the setting of linear functional data. The main contributions are the use of Bayesian Risk instead of an acquisition function and the implementation of SED as a Probabilistic Programming Language.

**Audience:**

Yes

**Claims And Evidence:**

No

**Requested Changes:**

- You should take into account the current literature, see e.g. https://arxiv.org/pdf/2302.14545.pdf. In particular cite literature for GPs being used in the application domain from the last sentence of the abstract. I will spare the authors with suggestions about what to cite and hiding my own papers in this list.
- Using # for the pushforward instead of the typical star seems weird to me.
- The notataion with $f$ and $\mathrm{f}$ in formulas is strange to me.
- The author(s) demand the linear functionals to be continuous. It seems to me that they should also be measurable w.r.t. some (not specified!) sigma-algebras. (Similar, you it might help to the topology on your function spaces, even though any reasonable choice seems fine on first glance.)
- In 3.3, why do you only learn two of the 4 parameters? Pleasse specify the reason.
- In F you specify that you approximate integrals in loss function by a finite Riemann sum. Did the design points prefer to be chosen at these grid points, as this would reduce the error locally even more? Can you comment on this? Is this effect even strong in F.4, where the points are far more spread out in 4D space?
- In E.3, the transform using logit might present you from sampling at the boundary of the design space, also due to a finite learning rate of stochastic optimizers. Can you comment on why this is not a problem for you?
- In Figure S11, I cannot quantify the difference between EI and BR. Consider incorporating a contour plot of an error between ground truth and the respective methods.
- Make and separete your contributions more clearly.

**Strengths And Weaknesses:**

Strengths
- The suggested PPL can deal with general nonlinear quantities of interest, reduces the specification of an acquisition function to the determination of a loss.
- The applications for this paper seem to very wide in general.
- The paper is well written and easy to follow.
- The additional experiments in the appendix are interesting

Weaknesses
- The suggested decision theoretic approach is only compared on 1 out of 4 experiments against a typical acquisition function (EI), otherwise random (2 times) or only validation on a senseful quantity. This is not sufficient to show the claims of the paper.
- The Literature is not up to date. The most recent literature is from 2021.
- The usage of Bayes risk in BO seems interesting. But from reading the paper, it is not really clear to me, whether this is a novelty.
- After reading the paper, it is not clear to me how P2 from the introduction is *really* answered. Instead of an acquisition function one needs to specify a loss function.
- The novelty feels strange. It is low when reading the introduction, but several minor interesting technical gems are hidden in the paper. Perhaps the paper needs a complete rewrite or restructuring? Is this a computational paper, where you present software? Or is it about demonstrating your approach of using a loss function to construct an acquisition function?
- It might have been interesting to see a loss function that is not a typical squared loss function.

---

> ### Comment · Reviewer_R11h · 2024-03-19
> **After Reading the other reviews**
>
> The review by Reviewer 3tPM makes similar comments as I make, and also raises some additional good points. I cannot recomment acceptance of the paper at TMLR at the current stage. I think a major rewrite is necessary, as some of the points raised by me and Reviewer 3tPM lie at the core of the contributions of the paper.

---

> > ### Comment · Action_Editor_JMov · 2024-03-19
> >
> > We are currently in the author discussion stage. The authors have the opportunity to respond to the first round of reviews before we can begin contemplating final recommendations.
> >
> > -AE

---

> ### Author Response · Authors · 2024-03-22
>
> Thank you very much for your careful reading of our manuscript:
>
> > Weaknesses
>
> > The suggested decision theoretic approach is only compared on 1 out of 4 experiments against a typical acquisition function (EI), otherwise random (2 times) or only validation on a senseful quantity. This is not sufficient to show the claims of the paper.
>
> Please allow us to emphasise we do not claim superiority of GaussED to any existing method on any particular task.  We simply wish to present the methodology and make the code available, and this is why we have submitted this manuscript to TMLR.  (The perceived significance of the manuscript is not an acceptance criterion for TMLR.)
>
> In fact, we would not expect to improve on acquisition functions that have been carefully designed by other researchers based on extensive theoretical and empirical investigations, as is the case in Bayesian optimisation, for example.  The claims we make in the manuscript are limited to (a) GaussED provides a simple high-level syntax for performing sequential experimental design with Gaussian processes and non-linear quantities of interest, and (b) GaussED does not require specification of an acquisition function, only specification of a loss function.  We believe these specific claims are supported by the evidence in the manuscript.
>
> All that being said, we did include an illustrative empirical comparison against the popular EI acquisition function for Bayesian optimisation in Appendix G3, where we observed that "expected improvement tends to behave more exploitatively, whereas Bayes risk tends to favour exploration".
>
> > The Literature is not up to date. The most recent literature is from 2021.
>
> Thank you for pointing this out; below we have responded to this point in more detail.
>
> > The usage of Bayes risk in BO seems interesting. But from reading the paper, it is not really clear to me, whether this is a novelty.
>
> Please note that we do not make any claim about whether our methodology carries any novelty for BO.  In fact, we have little interest in solving standard BO tasks, since many excellent methodologies already exist for BO.  Rather, our interest is in the development of software that can be brought to bear on a wider range of sequential experimental design problems, one canonical example of which is BO, but other examples include tomographic reconstruction, probabilistic solution of partial differential equations, and emulation of computer models.
>
> > After reading the paper, it is not clear to me how P2 from the introduction is really answered. Instead of an acquisition function one needs to specify a loss function.
>
> Thank you for the opportunity to expand on this point:  Recall that P2 stated our aim to "circumvent the requirement for the user to specify an acquisition function for SED [sequential experimental design]".  This is an important goal, since designing a good acquisition function for a new task is non-trivial.  Acquisition functions are usually designed for a specific application (e.g. Bayesian optimisation), and the search for a good acquisition function usually requires detailed theoretical and empirical work specific to the applied context.  (Indeed, the design of acquisition functions is still an active area of research in Bayesian optimisation.)  This is a problem for us, because we want GaussED to be application non-specific, and thus we cannot assume that an acquisition function is readily available - for example, in our tomographic example we are not aware of any particular acquisition function that has been proposed for this task.  How then to proceed?  In such circumstances, there will at least be a quantity of interest $q(\mathsf{f})$, and it would be reasonable to expect there to be an application-specific sense in which the quality of an approximation $\hat{q}$ to $q(\mathsf{f})$ can be measured.  With these two ingredients it is then possible to write down a loss function $L(\hat{q},q(\mathsf{f}))$, from which an acquisition function can be implicitly constructed - this is the approach taken in GaussED.  This rationale may not have come across clearly, and so we have made our presentation clearer in the revised manuscript.

---

> ### Author Response · Authors · 2024-03-22
>
> > The novelty feels strange. It is low when reading the introduction, but several minor interesting technical gems are hidden in the paper. Perhaps the paper needs a complete rewrite or restructuring? Is this a computational paper, where you present software? Or is it about demonstrating your approach of using a loss function to construct an acquisition function?
>
> Our aim, in submitting this manuscript, is simply to present the methodology and make the code available; this is why we have submitted this manuscript to TMLR.  As we are presenting methodology, computation, and software all in one paper, this brings with it some narrative challenges.  That being said, Reviewer 33Gt praised the presentation in the manuscript, and so we have elected not to undertake a major rewriting.  (A major rewriting is not really possible within the short revision period for TMLR.)
>
> > It might have been interesting to see a loss function that is not a typical squared loss function.
>
> This is an interesting point, because the squared loss function is more general than it first appears in our context:  Indeed, our default loss function is $L(\mathrm{g},\mathrm{g}') = \|q(\mathrm{g}) - q(\mathrm{g}')\|$ and through choosing $q(\cdot)$ we have quite a bit of control over what differences between $\mathrm{g}$ and $\mathrm{g}'$ are being measured.  For example, in the tomographic reconstruction example we picked $q(\mathrm{g}) = \exp(3\mathrm{g})$, which had the effect of focusing the loss function on the areas of the reconstruction associated with the highest densities.  This point has been clarified in the revised manuscript, and we are grateful that you have raised it.
>
> > Requested Changes:
>
> > You should take into account the current literature, see e.g. https://arxiv.org/pdf/2302.14545.pdf. In particular cite literature for GPs being used in the application domain from the last sentence of the abstract. I will spare the authors with suggestions about what to cite and hiding my own papers in this list.
>
> Thank you very much; we have cited this useful review article and updated our literature review with a representative selection of the references discussed in this review.
>
> > Using # for the pushforward instead of the typical star seems weird to me.
>
> Both $*$ and # are reasonably widely used; for example, both notations appear on https://en.wikipedia.org/wiki/Pushforward_measure.
>
> > The notataion with $f$ and $\mathsf{f}$ in formulas is strange to me.
>
> Thank you for the opportunity to address this point:  The purpose of this subtle notational distinction is to emphasise that $\mathsf{f}$ is the true data-generating function and $f$ is a dummy variable that takes values in an appropriate set of functions, ideally also containing $\mathsf{f}$.  In Bayesian statistics and machine learning we often overload notation and ignore this distinction (e.g. $\theta$ might be the parameter of a statistical model, and also a random variable representing uncertainty in the associated parameter), but in setting out the methodology for GaussED we felt it was important to (subtly) acknowledge this distinction, so that our methodology is rigorously presented.
>
> > The author(s) demand the linear functionals to be continuous. It seems to me that they should also be measurable w.r.t. some (not specified!) sigma-algebras. (Similar, you it might help to the topology on your function spaces, even though any reasonable choice seems fine on first glance.)
>
> Please note that in Appendix C1 we stated that the function space $\mathcal{F} = C^r(\mathcal{X})$ is the set of $r$-times continuously differentiable functions on $\mathcal{X}$ equipped with the norm $\max_{|\alpha| \leq r} \| f^{(\alpha)} \|_\infty$, and further that this normed space is endowed with the associated Borel sigma algebra generated by the open sets in $C^r(\mathcal{X})$.  Reviewer 33Gt praised the structure of the manuscript, which reserves these technical details for the appendices, and so we do not propose to move these details to the main text.
>
> > In 3.3, why do you only learn two of the 4 parameters? Pleasse specify the reason.
>
> This choice was made so that we can visualise the experimental designs in 2D (Figure 4).  There is no reason GaussED could not be applied to a 4D optimisation task, but (a) visualisation would be more difficult, and (b) for Lotka--Volterra there is a subset of the full 4D parameter space where the solution to the ODE explodes, which slightly complicates the analysis as we would need to introduce error handling into the ODE solver / likelihood.

---

> > ### Author Response · Authors · 2024-03-22
> >
> > > In F you specify that you approximate integrals in loss function by a finite Riemann sum. Did the design points prefer to be chosen at these grid points, as this would reduce the error locally even more? Can you comment on this? Is this effect even strong in F.4, where the points are far more spread out in 4D space?
> >
> > That is an interesting question; in 2D, where we can easily visualise results, the grid that we chose for the Riemann sum was relatively fine, so that the distance between grid points was of a similar order to the stochasticity that we see in the experimental design due to the use of stochastic optimisation.  The situation in 4D is less clear due to the difficulty of visualising the design, but we suspect that you are right.  We do not make any claims regarding this point in the manuscript, but we have additionally added a remark that this could occur if an insufficient number of cubature nodes are used.  (Of course, in an ideal world we would want to work with a sufficiently high number of cubature nodes that this issue is avoided.)
> >
> > > In E.3, the transform using logit might present you from sampling at the boundary of the design space, also due to a finite learning rate of stochastic optimizers. Can you comment on why this is not a problem for you?
> >
> > This is true - we make no claim about the optimality of the approach to stochastic optimisation that we have implemented.  On the other hand, global optimisation using stochastic gradients on constrained spaces is not a new problem, and more sophisticated approaches to optimisation could presumably be used in GaussED.  A remark to this effect has now been added to Appendix E.3 of the manuscript.  Thank you for raising this point.
> >
> > > In Figure S11, I cannot quantify the difference between EI and BR. Consider incorporating a contour plot of an error between ground truth and the respective methods.
> >
> > There are two differences to highlight:  (a) the difference between the designs produced by EI and BR, and (b) the difference between the reconstructed functions produced by EI and BR.
> >
> > For (a):  On the left hand side of the figure we see that the EI design contains clusters of design points, whereas BR does not.  This is explained in the manuscript as "expected improvement tends to behave more exploitatively, whereas Bayes risk tends to favour exploration".
> >
> > For (b):  We believe that visualisation of the reconstructed functions for EI and BR is better aligned with the aims of the paper, since we do not make any claim about which of these methods is best at reconstruction of the true data-generating function (they do a similar job, perhaps BR is slightly better in this example).
> >
> > > Make and separete your contributions more clearly.
> >
> > Thank you; we have clarified the main contributions by improving the text of the manuscript.  Please let us reiterate that no claims are made regarding the performance of GaussED compared to  existing methods on any particular task, so that, referring to the acceptance criteria for TMLR, the limited claims we do make are fully evidenced.

---

### Review · Reviewer_33Gt · 2024-03-18

**Summary Of Contributions:**

The authors design and implement an automated system for sequential experimental design. They use Gaussian process regression (GPs) to represent the true model ("quantity of interest") and expected negative Bayes risk as a catch-all acquistion function. Stochastic optimization with a reparametrization trick (for the GP) is used for design. They present promising results on several toy examples and finally on a "real" PDE model of the heart.

**Audience:**

Yes

**Claims And Evidence:**

Yes

**Requested Changes:**

See minor comments above

**Strengths And Weaknesses:**

The paper is very clearly written and the motivation is clear, with excessive math detail kept to the appendix. My only minor comments are:
- IMO it's confusing to talk about g being a decision rule (just after eq 4) when here it is just a parameter being estimated
- Kingma & Welling is not the original ref for the reparameterization trick, it was first used in Williams, R. J. Simple statistical gradient-following algorithms for connectionist reinforcement learning. Reinforcement learning 1992.
- Finally, I feel it's a bit of a stretch to call GaussED a PPL. Admittedly I'm not sure exactly what the bar is for this: maybe requiring a compiler or equivalent machinery (e.g. the tracing of pyro)?

Regardless, this is a well written paper providing a useful framework and software tool, which does not try to oversell its abilities. I recommend acceptance.

---

> ### Author Response · Authors · 2024-03-22
>
> Thank you for your careful reading of our manuscript:
>
> > IMO it's confusing to talk about $\mathrm{g}$ being a decision rule (just after eq 4) when here it is just a parameter being estimated
>
> Thank you, it would indeed be more accurate to call $\mathrm{g}$ the Bayes act.  This has been corrected.
>
> > Kingma & Welling is not the original ref for the reparameterization trick, it was first used in Williams, R. J. Simple statistical gradient-following algorithms for connectionist reinforcement learning. Reinforcement learning 1992.
>
> Thank you, the original reference has now been used.
>
> > Finally, I feel it's a bit of a stretch to call GaussED a PPL. Admittedly I'm not sure exactly what the bar is for this: maybe requiring a compiler or equivalent machinery (e.g. the tracing of pyro)?
>
> In retrospect we agree that the "probabilistic programming language" (PPL) terminology was not quite right.  We wanted to emphasise that our syntax is sufficiently high-level that the user does not need to engage explicitly with the operations of conditioning, Bayes' theorem, and so forth.  However, you are correct that GaussED can be considered a specialised tool rather than a general purpose PPL.  The name of the manuscript has been changed and the claim that GaussED is a PPL has been removed.

---

### Review · Reviewer_3tPM · 2024-03-19

**Summary Of Contributions:**

This paper introduces a framework for Bayesian optimization.  The contribution is angled in terms of probabilistic programming and sequential experimental design.  A link is made between the acquisition function and a Bayesian risk minimization formulation.  This circumvents directly specifying an acquisition function, in favor of an easier-to-specify loss function.  A GP is used as a latent function approximation family, allowing very straightforward inference and sampling.  Some empirical evaluation is presented and appears to perform well.

**Audience:**

Yes

**Broader Impact Concerns:**

No concerns.

**Claims And Evidence:**

No

**Requested Changes:**

Please answer the questions raised above, both in the review forum and make clarifications to these in the paper.

**Strengths And Weaknesses:**

# Review Summary

I think there is a good paper in this submission somewhere.  I have several high-level questions I have, which really limits how exactly I can evaluate the work.  I have tried to be overly thorough when listing them below in Questions.  Until then I err toward rejection, but am very willing to reevaluate this.

# Strengths
Methods combining AutoML and SED are clearly very appealing.  General methods for this are very appealing and would make these themes more broadly available.  Sidestepping specifying an acquisition function is nice, and using spectral approximations for the latent function is also nice.

The paper itself is relatively well written, with nicely prepared figures.  The main text itself is somewhat light on details, but the supplementary material is incredibly extensive, having a "textbook" feel in places.  I quite like how this is arranged so that the paper is direct, but more information is available.  There is some discussion of the computational effort required for these models, and code is also included.

I think overall the writing, pitch and level of the paper is correct for TMLR.

# Weaknesses
I tried to pose most of my weaknesses as questions below -- because I genuinely believe they might be misunderstandings on my part.  I do have some concrete weaknesses, however.

1.  *Empirical evaluation*:  The empirical evaluation is weak.  No baselines are really compared to, and there seems to be very little analysis of sensitivity to hyperparameters or comparison to different acquisition functions.

2. *Applicability*:   BayesOpt is a very general framework; but relying on a specific family of GPs does severely limit the number of problems this can be applied to (e.g. continuous problems).  This is okay, but is certainly a weakness.

3.  *Notational Density*:  As far as I can tell, you are specifying an acquisition function on a finite-dimensional parameter space.  I don't think the complexity of the notation used is really necessary.

# Questions.

1.  *Is this PP?*:  Foremost, I don't quite track how this is a probabilistic programming framework?  It seems to me like you're implicitly specifying a different acquisition function, and handing it off to an optimizer to get the next value.  This is totally fine on its own, but I am really struggling to place its relationship to wider PP methods.  The absence of discussion of control flow, permitted distributional families, samplers etc, and the limited-at-best evaluation (qualitative and quantitative) relative to other PP frameworks/languages is curious.  It seems to me more that this is a Bayesian optimization paper, focusing on a particular (implicit) parameterization of an acquisition function and latent function approximation family.  Can the authors please explain why this is a PP framework/language?

2.  *Ease of specifying q*:  The second ''contribution'' is circumventing the specification of an acquisition function (AF).  This sounds great, but really you are swapping specifying an AF with a $q$ function.  The authors assert that specifying a q is easier, but I don't really understand how.   EI, UCB etc are all well-studied methods with known strengths and weaknesses, and as such, i don't really have to ''specify'' an acquisition function, I just pick one.  In contrast, $q$ is something I have to design differently for every problem, so not only are you not gaining anything, you are actually adding a -- very important -- hyperparameter for which i have no intuition over.  For instance, how is $exp(3f)$ chosen?  There is remarkably little discussion of this.  It is very not clear to me that setting q is indeed easier.
    - This could be ameliorated by evaluating the sensitivity of the result to differences in q functions, vs differences in AFs.  As far as I can tell, the only comparison to any BayesOpt-like algorithm comes in the appendix (see further question below).

3.  *Intrinsic difference of Eq 4*:  Something I cannot wrap my head around is whether Equation 4 is fundamentally different from an acquisition function,  or whether it functions exactly the same as an acquisition function.  If this is true, the specifying through the q function actually obfuscates what the algorithm is optimizing?  Alternatively, is there an alternative interpretation of the newly defined acquisition function, in terms of EI, exploit explore, e.g. more conventional terms.  There is relatively little discussion of what Eq 4 is _actually doing_.  And crucially, what the net impact on SED is -- do you explore more or less?  Is it more robust to noise in the function approximation/optimization?  If the authors can comment on this, it might help me unlock the key benefit/delta here.

4.  *Further deltas*:  Looking at the pseudocode in Figure 1, I am querying if there is more to GaussED than specifying an (implicit) acquisition function and using a function approximator (spectral GP).  I would like to ask the authors to concretely and compactly re-state here any additional contributions of their work; because it does not shine through currently.

# Minor Questions/Weaknesses
- Basis functions are only concretely introduced in the supplement, despite being quite an important hyperparameter.
- In figure 2c, you say it is lower-bounded, but it clearly goes below the lower bound...?
- What is $z_i$ in the derivative term in the (unnumbered) equation after Eqn (7).
- I think the explanation of Experiment 3.2 is poor, and only single run is presented.  It also seems like the results are surprisngly poor, even when approximating what is quite a simple function.  It would also be nice if you highlighted _why_ certain red bands were selected at each step, as opposed to overlaying them all on at the same time -- does the function explore the space, or does it lock on first and then explore the space?
- Experiments examining the sensitivity to the number of basis functions should be in the main text.
- Citation style is incredibly inconsistent.

---

> ### Author Response · Authors · 2024-03-22
>
> Thank you very much for your careful reading of our manuscript:
>
> > Weaknesses
>
> The three weaknesses that you highlight are fair, and we do not strongly disagree with this assessment.  However, these points are not directly relevant to the acceptance criteria for TMLR; we simply wish to present the methodology and make the code available, and this is why we have submitted this manuscript to TMLR.  Indeed, we do not claim that GaussED is superior to existing solutions in any particular application domain, our claims are limited to (a) GaussED provides a simple high-level syntax for performing sequential experimental design with Gaussian processes and non-linear quantities of interest, and (b) GaussED does not require specification of an acquisition function, only specification of a loss function.
>
> > 3. Notational Density
>
> If we may briefly provide a counter-argument to your third perceived weakness, as it also refers to one of your questions below:  It is true that in some sense all numerical computation ends up being finite-dimensional, but the purpose of our set up and notation was to emphasise that there is a well-defined limit as the dimension of the underlying finite-dimensional representation is taken to infinity; increasing the number of basis functions used in our spectral approximation of the Gaussian process will eventually have a negligible effect.  This is important, because it ensures the choice of the number of spectral basis functions will not have a major impact on our scientific conclusions, provided that number is large enough.
>
> > Questions
>
> > 1. Is this PP?
>
> In retrospect we agree that the "probabilistic programming language" (PPL) terminology was not quite right.  We wanted to emphasise that our syntax is sufficiently high-level that the user does not need to engage explicitly with the operations of conditioning, Bayes' theorem, and so forth.  However, you are correct that GaussED can be considered a specialised tool rather than a general purpose PPL.  The name of the manuscript has been changed and the claim that GaussED is a PPL has been removed.
>
> > 2. Ease of specifying q
> > 3. Intrinsic difference of Eq. 4
>
> Thank you for the opportunity to clarify these two related points:  Acquisition functions are usually designed for a specific application (e.g. Bayesian optimisation), and the search for a good acquisition function usually requires detailed theoretical and empirical work specific to the applied context.  (Indeed, the design of acquisition functions is still an active area of research in Bayesian optimisation.)  This is a problem for us, because we want GaussED to be application non-specific, and thus we cannot assume that an acquisition function is readily available - for example, in our tomographic example we are not aware of any particular acquisition function that has been proposed for this task.  How then to proceed?  In such circumstances, there will at least be a quantity of interest $q(\mathsf{f})$, and it would be reasonable to expect there to be an application-specific sense in which the quality of an approximation $\hat{q}$ to $q(\mathsf{f})$ can be measured.  With these two ingredients it is then possible to write down a loss function $L(\hat{q},q(\mathsf{f}))$, from which an acquisition function can be implicitly constructed - this is the approach taken in GaussED.  This rationale may not have come across clearly, and so we have made our presentation clearer in the revised manuscript.
>
> > 4. Further deltas
>
> Your summary here is accurate; our contribution is limited to setting out this methodology and providing code to implement it - this is why we have submitted the manuscript to TMLR.  The motivation of GaussED is to be application non-specific, and this includes handling non-linear quantities of interest related to the GP.  For example, the breadth of applicability of GaussED includes Bayesian optimisation, tomographic reconstruction, probabilistic solution of PDEs, and emulation, which together showcase the applicability of GaussED.

---

> > ### Author Response · Authors · 2024-03-22
> >
> > > Minor Questions / Weaknesses
> >
> > > Basis functions are only concretely introduced in the supplement, despite being quite an important hyperparameter.
> > > Experiments examining the sensitivity to the number of basis functions should be in the main text.
> >
> > Following on from our response to "Notational Density", we note that the output of GaussED converges as the number of basis functions increases, so in principle there is arbitrarily little sensitivity to this hyperparameter provided it is taken large enough.  A legitimate question is then how large this parameter needs to be taken, and we discuss this (with experiments) in Appendix G2.  The current division of material between main text and appendix was praised by Reviewer 33Gt, and we have elected to maintain this split.
> >
> > > In figure 2c, you say it is lower-bounded, but it clearly goes below the lower bound...?
> >
> > The blue line is here to indicate a slope of -(1/2) for the purpose of visually interpreting convergence rates, and is not an explicit bound.  This has been made clear in the manuscript.
> >
> > > What is $z_i$ in the derivative term in the (unnumbered) equation after Eqn (7).
> >
> > Here $z$ is a vector in $\mathbb{R}^m$, so $z_i$ is the $i$th coordinate of this vector.
> >
> > > I think the explanation of Experiment 3.2 is poor, and only single run is presented. It also seems like the results are surprisngly poor, even when approximating what is quite a simple function. It would also be nice if you highlighted why certain red bands were selected at each step, as opposed to overlaying them all on at the same time -- does the function explore the space, or does it lock on first and then explore the space?
> >
> > Referring to the acceptance criteria or TMLR, please note that no claim is made about the quality of the results, except that they are superior to using a random design for the tomographic reconstruction - and this limited claim is evidenced.  The top panel of Figure 3 indicates how the experimental design was sequentially constructed, and we would like to briefly explain why this was the cleanest visualisation that we could come up with:  Recall that for each experiment we are effectively selecting a line in 2D, and that a line is described by an intercept and a slope.  We could in principle plot the optimisation landscape in (intercept,slope) space, but we thought this would be hard to interpret since it is in a different coordinate system to the object which we are aiming to reconstruct.
> >
> > > Citation style is incredibly inconsistent.
> >
> > Thank you for pointing this out, which was an oversight on our part during conversion to the TMLR format.  This has been fixed.

---

### Decision · Action_Editor_JMov · 2024-04-10

**Recommendation:** Reject

**Comment:**

This manuscript generated a great deal of discussion regarding the criteria for acceptance for TMLR and their proper interpretation and implementation. I have elaborated on both of the primary acceptance criteria and my interpretation of the situation in light of the reviews, discussion, and recommendations above. To summarize:
- Regarding claims and evidence, the authors limited their claims to "simply [presenting] the methodology and [making] the code available." However, even this reduced claim must be supported by accurate, convincing and clear evidence, and a majority of the reviewers felt that the authors were not successful in doing so.
- Regarding target audience, a natural consequence of limiting the claims made in the paper is limiting the number of interested readers. A majority of the reviewers felt that the relatively narrow scope of the paper may substantially reduce the size of its target audience to the point where "the spirit of the law" may be violated here. However, I do believe that addressing the "claims and evidence" issues raised by the reviewers would likely satisfy the "target audience" issues as well.

I want to stress that I believe this manuscript could be suitably revised to address the concerns of reviewers R11h and 3tPM. Indeed, the authors seem to agree that the manuscript could be significantly strengthened, but questioned their ability to make major revisions on the short timescale enforced by TMLR. I would like to strongly encourage them to submit a suitably revised manuscript to TMLR for reconsideration if they are so inclined.

**Audience:**

The overarching topic of the submission -- sequential experimental design -- is certainly of interest to a nontrivial subset of TMLR's audience. However, two reviewers expressed serious concerns regarding the structure and scope of the paper and the chosen evaluation methodology. Although these are not explicitly mentioned in TMLR's acceptance criteria, the reviewers argued that these issues may substantially limit the submitted manuscript's interest to the TMLR audience.

**Claims And Evidence:**

There was not universal agreement among the reviewers on this front. However, a majority of the reviewers argued that the manuscript in fact makes such limited claims regarding its contributions that there's relatively little substance to support in the first place.

A major point of contention appears to have been regarding the scope of the empirical evaluation (or relative lack thereof), which is one primary mechanism (although not the only such mechanism) to support claims made in papers submitted to and published in TMLR. The authors argue that TMLR allows authors to, in lieu of running more experiments, simply reduce the scope of their claims. The suitably reduced claim proposed by the authors in response is
> simply to present the methodology and make the code available.

However, TMLR requires that even this be supported by accurate, _convincing_ and clear evidence. A software package with exemplary design on paper is of no use if the implementation is poor or otherwise yields poor results. That's especially the case for many of the target use cases for the software in question, where evaluating the system of interest may be extremely expensive. This situation would compel a rational practitioner to be extraordinarily risk averse in selecting a software package for driving an experimental design process. The consensus among the reviewers seems to be that the submitted manuscript does not convincingly support the central software contribution, e.g., in terms of software/implementation quality. Note that this limitation has implications for the audience question below as well.

**Resubmission Of Major Revision:**

The authors may consider submitting a major revision at a later time.